# Synergistic stabilization of microtubules by BUB-1, HCP-1, and CLS-2 controls microtubule pausing and meiotic spindle assembly

Nicolas Macaisne[1†], Laura Bellutti[1†], Kimberley Laband[1†], Frances Edwards[1†], Laras Pitayu-Nugroho[1], Alison Gervais[1], Thadshagine Ganeswaran[1], Hélène Geoffroy[1], Gilliane Maton[1], Julie C Canman[2], Benjamin Lacroix[3]*, Julien Dumont[1]*

[1]Université Paris Cité, CNRS, Institut Jacques Monod, F-75013, Paris, France; [2]Columbia University; Department of Pathology and Cell Biology, New York, United States; [3]Centre de Recherche en Biologie Cellulaire de Montpellier (CRBM), CNRS UMR 5237, Université de Montpellier, Montpellier, France

*For correspondence:
benjamin.lacroix@crbm.cnrs.fr
(BL);
julien.dumont@ijm.fr (JD)

†These authors contributed equally to this work

Competing interest: The authors declare that no competing interests exist.

**Abstract** During cell division, chromosome segregation is orchestrated by a microtubule-based spindle. Interaction between spindle microtubules and kinetochores is central to the bi-orientation of chromosomes. Initially dynamic to allow spindle assembly and kinetochore attachments, which is essential for chromosome alignment, microtubules are eventually stabilized for efficient segregation of sister chromatids and homologous chromosomes during mitosis and meiosis I, respectively. Therefore, the precise control of microtubule dynamics is of utmost importance during mitosis and meiosis. Here, we study the assembly and role of a kinetochore module, comprised of the kinase BUB-1, the two redundant CENP-F orthologs HCP-1/2, and the CLASP family member CLS-2 (hereafter termed the BHC module), in the control of microtubule dynamics in *Caenorhabditis elegans* oocytes. Using a combination of in vivo structure-function analyses of BHC components and in vitro microtubule-based assays, we show that BHC components stabilize microtubules, which is essential for meiotic spindle formation and accurate chromosome segregation. Overall, our results show that BUB-1 and HCP-1/2 do not only act as targeting components for CLS-2 at kinetochores, but also synergistically control kinetochore-microtubule dynamics by promoting microtubule pause. Together, our results suggest that BUB-1 and HCP-1/2 actively participate in the control of kinetochore-microtubule dynamics in the context of an intact BHC module to promote spindle assembly and accurate chromosome segregation in meiosis.

## Editor's evaluation

This paper on the regulation of microtubule dynamics during *C. elegans* meiosis presents important findings that will be of interest to scientists in the broad field of microtubule function in both mitosis and meiosis. The experiments are beautifully conducted and presented and support the conclusions of the paper in a compelling manner. The results are interesting and add to our understanding of the control of microtubule dynamics at the kinetochore and its functional consequences for meiosis.

## Introduction

Equal partitioning of the replicated genome between the two daughter cells is a key step of cell division. Throughout meiosis and mitosis, proper interactions between spindle microtubules and kinetochores, multiprotein complexes assembled at the centromeres of meiotic and mitotic chromosomes, are essential for accurate chromosome segregation (*Musacchio and Desai, 2017*). Kinetochore-microtubule attachments are required for co-orientation of homologous chromosomes attached to the same spindle pole during meiosis I and for chromosome bi-orientation with sister chromatids attached to microtubules emanating from opposite spindle poles in meiosis II and mitosis (*Dumont and Desai, 2012*).

After nuclear envelope breakdown (NEBD), dynamic microtubules grow toward the chromosomes where they engage in lateral interactions with kinetochore-localized motor proteins (*Rieder and Alexander, 1990*; *Scaërou et al., 1999*; *Yao et al., 1997*; *Kapoor et al., 2006*; *Ferreira and Maiato, 2021*; *Renda et al., 2022*). These initial interactions promote chromosome orientation and accelerate stable end-on attachments with kinetochore-microtubules mediated by the Ndc80 complex (*DeLuca et al., 2006*; *Cheeseman et al., 2006*; *Wei et al., 2007*; *Ciferri et al., 2008*; *Cheerambathur et al., 2013*). Initially spindle microtubules are highly dynamic to allow spindle assembly and capture by kinetochores, which is essential for chromosome alignment. But as meiosis and mitosis progress, kinetochore microtubules become stabilized for efficient segregation of homologous chromosomes and sister chromatids (*Maia et al., 2012*; *Kabeche and Compton, 2013*; *Dumitru et al., 2017*). Thus, precise control of microtubule dynamics is essential for spindle assembly and the stepwise attachment of chromosomes followed by their accurate segregation.

Proteins of the cytoplasmic linker-associated protein (CLASP) family are evolutionary-conserved regulators of microtubule dynamics (*Akhmanova et al., 2001*; *Akhmanova and Steinmetz, 2010*; *Lawrence et al., 2020*). During meiosis and mitosis, CLASP proteins prevent spindle abnormalities and chromosome segregation errors in most species including yeast, *Drosophila*, *C. elegans* and mammals (*Pasqualone and Huffaker, 1994*; *Lemos et al., 2000*; *Cheeseman et al., 2005*; *Maiato et al., 2002*). In vitro, CLASPs maintain microtubules in a growing state by promoting microtubule rescue while inhibiting catastrophe (*Al-Bassam et al., 2010*; *Yu et al., 2016*; *Moriwaki and Goshima, 2016*; *Lawrence et al., 2018*; *Aher et al., 2018*). In dividing human cells, two paralogous CLASP1/2 proteins act redundantly at the kinetochore where they are targeted through their C-terminal domain (CTD) by a poorly characterized pathway that involves the motor protein CENP-E and the kinetochore and spindle-associated protein SPAG5/Astrin (*Maiato et al., 2003*; *Pereira et al., 2006*; *Maffini et al., 2009*; *Manning et al., 2010*; *Kern et al., 2016*). In *C. elegans*, CLS-2 is the sole CLASP ortholog that localizes at the kinetochore and is essential for normal spindle assembly and chromosome segregation (*Cheeseman et al., 2005*). During meiosis in *C. elegans* oocytes, CLS-2 is essential for meiotic spindle assembly, chromosome segregation and polar body extrusion (*Dumont et al., 2010*; *Laband et al., 2017*; *Pelisch et al., 2019*; *Schlientz et al., 2020*). Kinetochore localization of CLS-2 requires interaction with the two CENP-F-like proteins HCP-1/2, which are themselves localized downstream of BUB-1 (*Figure 1A*; *Cheeseman et al., 2005*; *Essex et al., 2009*; *Edwards et al., 2018*).

Bub1 (BUB-1 in *C. elegans*) is a kinase originally identified for its role in the Spindle Assembly Checkpoint (SAC), a safety mechanism that ensures proper connection of kinetochores to spindle microtubules (*Lara-Gonzalez et al., 2021*; *Zhang et al., 2022*). During meiosis and mitosis, Bub1 is also directly involved in chromosome bi-orientation through its non-SAC functions that: (1) promote kinetochore recruitment of dynein and CENP-F, (2) ensure proper inner-centromere localization of Aurora B, (3) recruit PP2A:B56 on meiotic chromosomes, and (4) limit kinetochore-microtubule attachment maturation by the SKA complex in mitosis (*Essex et al., 2009*; *Edwards et al., 2018*; *Johnson et al., 2004*; *Klebig et al., 2009*; *Kawashima et al., 2010*; *Zhang et al., 2015*; *Ciossani et al., 2018*; *Berto et al., 2018*; *Bel Borja et al., 2020*). During mitosis, Bub1 interacts physically with Bub3 via its 'Bub3-binding motif', formerly known as the GLEBS domain (*Wang et al., 2001*; *Larsen et al., 2007*; *Primorac et al., 2013*). The Bub1/Bub3 complex is then recruited to kinetochores through Bub3 direct-binding to phosphorylated MELT (Met-Glu-Leu-Thr) repeats located in the N-terminal half of Knl1 (*Primorac et al., 2013*; *Cheeseman et al., 2004*; *Shepperd et al., 2012*; *London et al., 2012*; *Yamagishi et al., 2012*; *Vleugel et al., 2013*). During meiosis in *C. elegans* oocytes, BUB-1 also localizes to kinetochores, which display characteristic cup-like shapes (*Monen et al., 2005*). This kinetochore localization requires KNL-1, but whether it occurs via BUB-3 and the KNL-1 MELT repeats

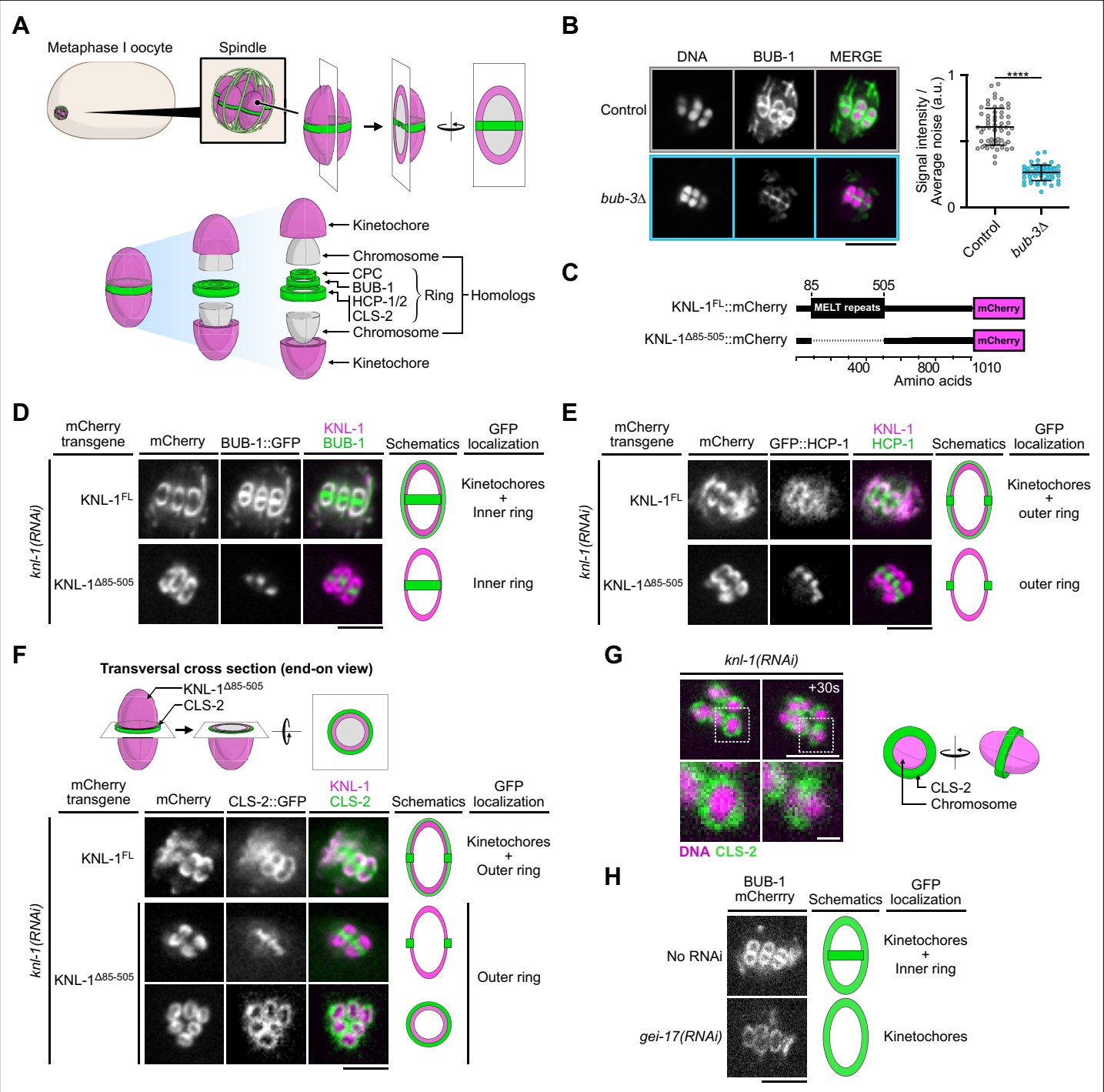

**Figure 1.** The N-terminal MELT repeats of KNL-1, but not BUB-3, are required for BHC module kinetochore targeting in oocytes. (**A**) Schematic of kinetochore and ring domain protein organization around a bivalent chromosome during metaphase I in a *C. elegans* oocyte. CPC: Chromosome Passenger Complex. (**B**) Immunolocalization of BUB-1 (left) and quantification of BUB-1 signal at kinetochores (right) in *bub-3(ok3437)* mutants (*bub3Δ*, n=58) compared to wild type controls (n=58). Error bars, Mean and standard deviation. Unpaired t-test, alpha = 0.05, p<0.0001. (**C**) Schematic of KNL-1::mCherry protein fusions. (**D–F**) Localization of BUB-1::GFP (**D**), GFP::HCP-1 (**E**) and CLS-2::GFP (**F**) in worms carrying full length or MELT-deleted KNL-1::mCherry (KNL-1$^{FL}$ and KNL-1$^{Δ85-505}$ respectively, n≥10). (**G**) Localization of CLS-2::GFP at ring domains in *knl-1*-depleted oocytes (left) with corresponding schematic (right). (**H**) Localization of BUB-1::mCherry in *gei-17*-depleted oocytes (n=29) compared to controls (n=25). Scale bars 5 μm, 1 μm in insets.

The online version of this article includes the following source data and figure supplement(s) for figure 1:

**Source data 1.** Panel B source data.

*Figure 1 continued on next page*

*Figure 1 continued*

**Figure supplement 1.** BUB-3 is not required for BHC module kinetochore targeting.

**Figure supplement 1—source data 1.** Panel A source data 1.

**Figure supplement 1—source data 2.** Panel A source data 2.

**Figure supplement 1—source data 3.** Panel B source data.

**Figure supplement 1—source data 4.** Panel C source data.

is unknown (*Dumont et al., 2010*). BUB-1 additionally concentrates at trilaminar ring domains located between each pair of homologous chromosomes in meiosis I and between sisters in meiosis II (*Dumont et al., 2010*). This localization lies downstream of the Chromosomal Passenger Complex (CPC) and requires BUB-1 sumoylation (*Pelisch et al., 2019*; *Pelisch et al., 2017*; *Davis-Roca et al., 2018*). The specific functions of these two distinct chromosomal BUB-1 localizations are at present unclear. However, BUB-1 plays a critical role during oocyte meiosis as its depletion leads to severe chromosome segregation errors and meiotic spindle abnormalities (*Dumont et al., 2010*). These meiotic phenotypes have been attributed to the role of BUB-1 in the recruitment of HCP-1/2 and CLS-2 at kinetochores and ring domains, the SUMO-dependent targeting of the chromokinesin KLP-19, and the phospho-dependent recruitment of PP2A:B56 to ring domains (*Dumont et al., 2010*; *Bel Borja et al., 2020*; *Davis-Roca et al., 2018*; *Wignall and Villeneuve, 2009*). However, the exact function of BUB-1 during meiosis in oocytes is unclear.

In mammals, CENP-F is a large coiled-coil protein recruited to kinetochores through the physical interaction between a specific targeting domain and the kinase domain of Bub1 (*Ciossani et al., 2018*; *Liao et al., 1995*). CENP-F contains two high-affinity microtubule binding domains (MTBDs), located at either terminus of the protein, required for the generation of normal interkinetochore tension and stable kinetochore-microtubule attachments (*Feng et al., 2006*; *Musinipally et al., 2013*; *Volkov et al., 2015*; *Kanfer et al., 2017*). CENP-F is also involved in the recruitment of the dynein motor at kinetochores via its direct interaction with the NudE/L dynein adaptor proteins (*Faulkner et al., 2000*; *Tai et al., 2002*; *Liang et al., 2007*; *Vergnolle and Taylor, 2007*; *Simões et al., 2018*). Despite these contributions to the process of chromosome alignment, CENP-F is non-essential in mammals as evidenced by the lack of segregation defects in CRISPR knockouts in human cells and by the viability of CENP-F knockout mice (*Pfaltzgraff et al., 2016*; *McKinley and Cheeseman, 2017*; *Raaijmakers et al., 2018*). In contrast in *C. elegans*, the two CENP-F-like proteins HCP-1/2 are essential for embryonic viability (*Cheeseman et al., 2005*). Together with their downstream partner CLS-2, kinetochore-localized HCP-1/2 control kinetochore-microtubule dynamics to prevent sister chromatid co-segregation to the same spindle pole in mitosis, and promote midzone microtubule assembly for central spindle formation in anaphase (*Cheeseman et al., 2005*; *Maton et al., 2015*; *Edwards et al., 2015*; *Hirsch et al., 2022*). During meiosis in *C. elegans* oocytes, HCP-1/2 and CLS-2 localize along spindle microtubules, at the cup-shaped kinetochores and on the ring domains (*Dumont et al., 2010*; *Laband et al., 2017*). Together they are essential for the formation of bipolar meiotic acentrosomal spindles in oocytes, for meiotic chromosome segregation and for efficient cytokinesis during polar body extrusion (*Dumont et al., 2010*; *Laband et al., 2017*; *Pelisch et al., 2019*; *Schlientz et al., 2020*). Therefore, BUB-1, HCP-1/2, and CLS-2, hereafter termed the BHC module, form a kinetochore module essential for meiosis and mitosis. How is this kinetochore module assembled and whether BUB-1 and HCP-1/2 only act as CLS-2 kinetochore-targeting subunits or whether they participate in regulating kinetochore microtubule dynamics is unknown.

In this study, we investigated the assembly and function of the BHC kinetochore module in the *C. elegans* oocyte and one-cell embryo. Through a combination of genetic approaches and live cell imaging, we found that the BUB-1 kinase domain, specific C- and N-terminal sequences of HCP-1, and the CLS-2 CTD are essential for BHC module assembly. Our in vitro TIRF (Total Internal Reflection Microscopy)-based microtubule assays demonstrated that BHC module components decreased the catastrophe frequency, while increasing the growth rate and rescue frequency. Surprisingly, they also induced a strong synergistic increase of the time spent in pause by microtubules. Overall, our results suggest that BUB-1 and HCP-1/2 are not mere targeting components for CLS-2 at the kinetochore, but instead actively participate in the control of kinetochore-microtubule dynamics in the context of an intact BHC module, which is essential for accurate chromosome segregation.

## Results

### The N-terminal MELT repeats of KNL-1, but not BUB-3, are essential for BHC module kinetochore targeting in oocytes (Figure 1)

We first determined the molecular mechanisms that target the BHC module to kinetochores and ring domains in *C. elegans* oocytes. In most systems during mitosis, Bub1 localization to kinetochores requires a physical interaction with Bub3, which in turn interacts with the phosphorylated MELT repeats of KNL-1 (*Primorac et al., 2013*; *Cheeseman et al., 2004*; *Shepperd et al., 2012*; *London et al., 2012*; *Yamagishi et al., 2012*; *Vleugel et al., 2013*; *Espeut et al., 2015*). We first analyzed if this was the case in *C. elegans* by analyzing a viable *bub-3* deletion mutant (*bub-3(ok3437)*, referred to as *bub-3Δ*). As previously shown, BUB-1 is destabilized in this mutant with its overall protein levels down to below 25% of the levels in control worms (*Figure 1—figure supplement 1A*) *Kim et al., 2015*. Accordingly, BUB-1 was also significantly reduced at meiotic kinetochores in this mutant, albeit to a much lower extent (57% reduction on average compared to controls) than the overall protein level reduction (*Figure 1B*). This surprisingly suggested that, in contrast to in yeasts or mammalian cells, BUB-1 can be recruited at kinetochores independently of BUB-3 in *C. elegans* oocytes. Importantly, the reduced BUB-1 kinetochore level in the *bub-3* deletion mutant, correlated with significant reductions of HCP-1 (36% reduction) and CLS-2 (58% reduction) at meiotic kinetochores (*Figure 1—figure supplement 1B–C*). This suggests that in *C. elegans* oocytes the BHC module can be recruited independently of BUB-3 to kinetochores.

We next tested if the KNL-1 MELT repeats are required for BUB-1 kinetochore targeting in meiosis by using a *C. elegans* strain expressing a truncation mutant of KNL-1 (KNL-1$^{\Delta85-505}$) that lacks all MELT repeats (*Kim et al., 2015*; *Moyle et al., 2014*; *Figure 1C*). We analyzed the localization of GFP-tagged BUB-1 in strains expressing RNAi-resistant transgenes encoding full-length KNL-1 (KNL-1$^{FL}$) or KNL-1$^{\Delta85-505}$ after depletion of endogenous KNL-1 by RNAi treatment. As expected, in the presence of KNL-1$^{FL}$, BUB-1 localized to cup-shaped kinetochores and to ring domains. In contrast in the presence of KNL-1$^{\Delta85-505}$, BUB-1 failed to localize to kinetochores (*Figure 1D*). GFP-tagged HCP-1 and CLS-2 also did not localize to kinetochores in KNL-1$^{\Delta85-505}$ (*Figure 1E and F*). The remaining chromosomal signal of GFP-tagged HCP-1 and CLS-2 in this mutant corresponded to their KNL-1-independent ring domain localization, as evidenced by the identical GFP pattern observed in KNL-1-depleted oocytes (*Figure 1G*). Thus kinetochore, but not ring-domain, localization of BUB-1 depends on the KNL-1 MELT repeats. In line with a previous study, we also confirmed that BUB-1 ring domain localization required the E3 SUMO-protein ligase GEI-17, and thus probably BUB-1 sumoylation (*Figure 1H*; *Pelisch et al., 2017*). These results show that the BHC module is recruited to kinetochores via the KNL-1 MELT repeats.

### Molecular determinants of BHC module assembly (Figure 2)

Next, we analyzed the domains of BUB-1, HCP-1, and CLS-2 necessary for BHC module assembly. During mitosis, CENP-F in mammals and HCP-1 in *C. elegans*, are recruited to kinetochores through the BUB-1 kinase domain (*Edwards et al., 2018*; *Ciossani et al., 2018*). We first tested if that was also the case during meiosis by analyzing HCP-1 and CLS-2 localization in *C. elegans* oocytes expressing RNAi-resistant transgenes encoding full length BUB-1 (BUB-1$^{FL}$) or a kinase domain-deleted mutant of BUB-1 (BUB-1$^{\Delta KD}$) after depletion of endogenous BUB-1 (*Figure 2A*; *Edwards et al., 2018*). GFP-tagged HCP-1 (and CLS-2) localized to kinetochores and ring domains in the presence of BUB-1$^{FL}$, but not BUB-1$^{\Delta KD}$ (*Figure 2B and C*). To determine if the kinase activity of BUB-1 was required for HCP-1 and CLS-2 kinetochore and ring targeting, we also analyzed their localizations in oocytes expressing a kinase dead (BUB-1$^{D814N}$) version of BUB-1 (*Moyle et al., 2014*; *Figure 2A*). Upon depletion of endogenous BUB-1 in the presence of the kinase-dead BUB-1$^{D814N}$ mutant, HCP-1 (and CLS-2) were normally targeted to kinetochores and rings (*Figure 2B and C*). Therefore, BUB-1 recruits HCP-1 (and CLS-2) to kinetochores and rings through its kinase domain, but independently of kinase activity, in meiosis in *C. elegans*.

In mammals, CENP-F kinetochore targeting involves a physical interaction between the specific C-terminal CENP-F targeting domain and the kinase domain of Bub1 (*Ciossani et al., 2018*). HCP-1 and 2 are long coiled-coil proteins that only share 27.5% and 24.7% similarity respectively with CENP-F, which makes identification of conserved functional domains more challenging (*Hirsch et al., 2022*). To identify the domains of HCP-1 responsible for its localization, we thus generated a series of transgenic

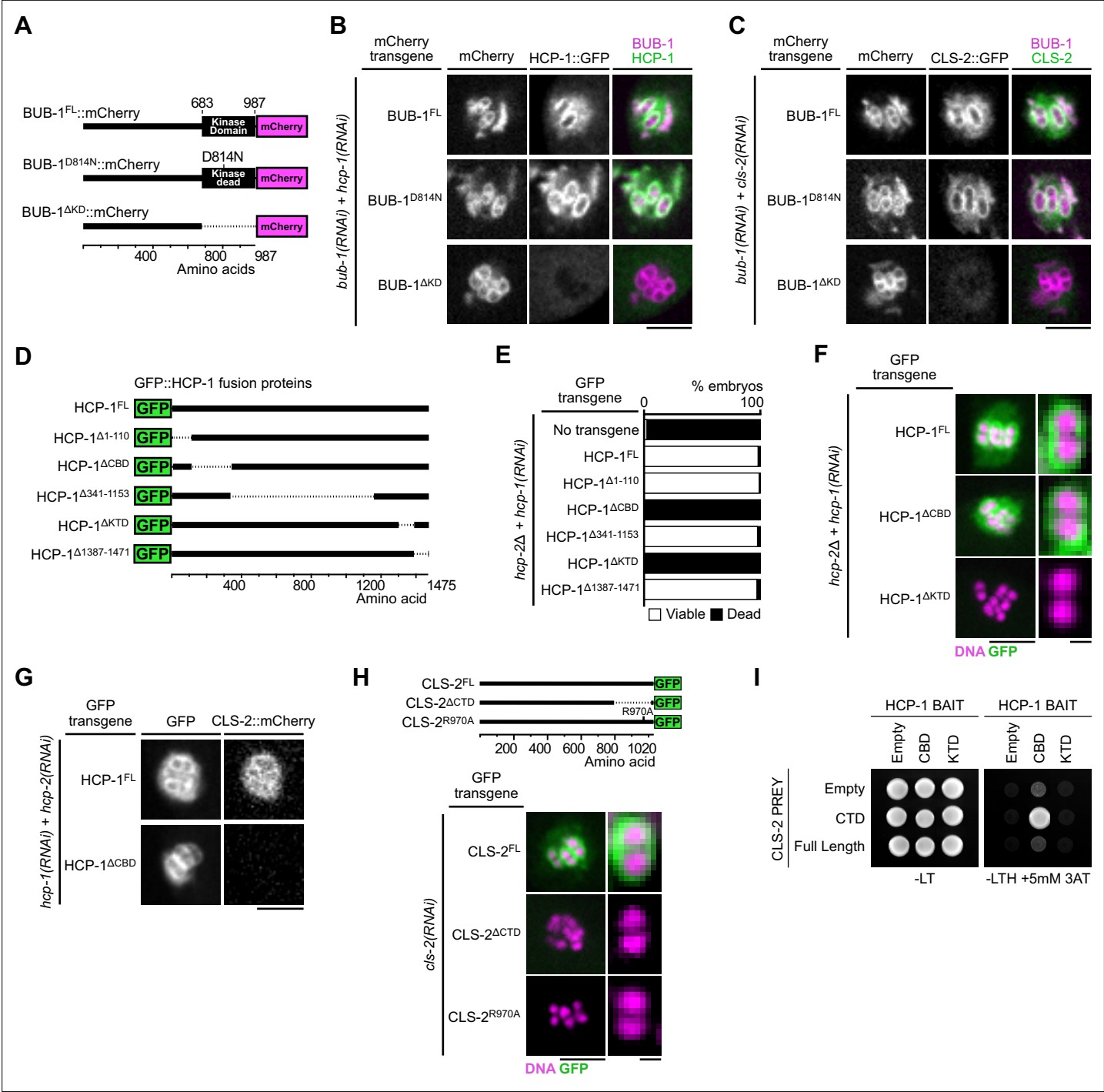

**Figure 2.** Molecular determinants of BHC module assembly. (**A**) Schematic of BUB-1::mCherry protein fusions. (**B,C**) Localization of indicated RNAi-resistant BUB-1::mCherry and GFP::HCP-1 (**B**) or CLS-2::GFP (**C**) upon depletion of corresponding endogenous gene products (n≥10). (**D**) Schematic of truncated GFP::HCP-1 fusions. (**E**) Embryonic viability assay in indicated transgenic *hcp-2(ijm6)* (*hcp-2Δ*) mutants upon depletion of endogenous *hcp-1*. (**F**) Localization of indicated GFP::HCP-1 fusions in *hcp-2Δ* worms depleted of endogenous *hcp-1* (n≥9). (**G**) Localization of CLS-2::mCherry in indicated conditions (n≥13). (**H**) Localization of schematized (top) RNAi-resistant CLS-2::GFP fusions upon depletion of endogenous *cls-2* (bottom, n≥10). (**I**) Yeast-two-hybrid interaction assay between HCP-1 domains (baits) and CLS-2 CTD (prey). Scale bars, metaphase plate 5 µm, single chromosome details 1 µm.

The online version of this article includes the following source data and figure supplement(s) for figure 2:

**Source data 1.** Panel E source data.

**Figure supplement 1.** Protein domains essential for BHC module assembly.

*Figure 2 continued on next page*

*Figure 2 continued*

**Figure supplement 1—source data 1.** Panel B source data.

**Figure supplement 1—source data 2.** Panel C source data.

**Figure supplement 2.** Protein sequence alignment of indicated eukaryotic CLASP CTDs.

strains expressing RNAi-resistant GFP-fused truncations of HCP-1 (*Figure 2D*, *Figure 2—figure supplement 1A*). We focused on HCP-1 for this work because, although HCP-1 and 2 are functionally redundant and can compensate for each other to support embryonic development, HCP-1 plays a more primary role in early embryos (*Hirsch et al., 2022*). We introduced the HCP-1 transgenes in an *hcp-2* deletion mutant (hereafter *hcp-2Δ*) and analyzed embryonic viability after endogenous HCP-1 depletion by RNAi (*Edwards et al., 2018*). We first verified that GFP-fused wild-type HCP-1 could rescue embryonic lethality in absence of endogenous HCP-1/2. Two main regions of HCP-1 (amino acid 111–340 and 1311–1386, hereafter referred to as CBD (CLS-2 Binding Domain) and KTD (Kinetochore Targeting Domain) respectively, see below), one at either terminus, were essential for embryonic viability (*Figure 2E*, *Figure 2—figure supplement 1B*), even though the mutant and wild-type transgenes were expressed at comparable levels (*Figure 2—figure supplement 1C*). Although this domain conformation is reminiscent of the 2 MTBDs of human CENP-F, we could not identify significant sequence or structural similarity between the human CENP-F and *C. elegans* HCP-1 domains (*Volkov et al., 2015*).

We next tested if these C- and N-terminal domains in HCP-1 could instead be important for kinetochore localization of the corresponding GFP-fused truncated proteins. We found that deleting the KTD (HCP-1$^{\Delta KTD}$), but not the CBD (HCP-1$^{\Delta CBD}$), prevented kinetochore targeting of the corresponding deletion mutant HCP-1 (*Figure 2F*). The KTD could be responsible for kinetochore targeting via binding to the kinase domain of BUB-1, although we were unable to confirm direct interaction with a yeast-two-hybrid assay between the HCP-1 KTD, or a larger domain containing the KTD (1154–1386), and full-length BUB-1 or the BUB-1 kinase domain (*Figure 2—figure supplement 1D*). As the CBD is not required for HCP-1 kinetochore targeting, embryonic lethality in the corresponding mutant could instead be caused by a defect in CLS-2 recruitment to kinetochores. Accordingly, we found that the HCP-1 CBD truncation mutant prevented mCherry-fused CLS-2 kinetochore targeting (*Figure 2G*, *Figure 2—figure supplement 1E*). We also found identical results for HCP-1 and CLS-2 localizations in mitosis (*Figure 2—figure supplement 1F*). Thus HCP-1 kinetochore targeting requires its KTD, while the HCP-1 CBD mediates CLS-2 kinetochore recruitment.

We then sought to identify the CLS-2 domain that interacts with the HCP-1 CBD and is essential for its kinetochore targeting. In most species, CLASPs targeting to their various subcellular localizations, including to kinetochores, relies on interactions with various adapter proteins via a C-terminal domain (CTD; *Akhmanova et al., 2001*; *Maiato et al., 2003*; *Hannak and Heald, 2006*; *Lansbergen et al., 2006*). To test the function of the CLS-2 CTD, we generated a transgenic strain expressing a GFP-tagged deletion of the CTD (CLS-2$^{\Delta CTD}$). We compared the localization of the corresponding protein to that of full length GFP-fused CLS-2 (CLS-2$^{FL}$) following depletion of endogenous CLS-2. Both transgenes were expressed at similar levels (*Figure 5—figure supplement 1D*). In contrast to CLS-2$^{FL}$, the CTD-deleted transgenic protein did not target to the kinetochores during meiosis in oocytes or in mitosis in zygotes (*Figure 2H*, *Figure 5—figure supplement 1E*), which is in line with previous findings on vertebrate CLASPs (*Maiato et al., 2003*). Moreover, by comparing the CLASP CTD sequences of various species, we identified few, but strongly, conserved residues, including a stretch of three amino acids comprised of a Valine, an Arginine and a Lysine (VRK) (*Figure 2—figure supplement 2*). To determine the functional significance of this conservation, we mutated the central Arginine of the conserved VRK motif to an Alanine, and we generated the corresponding transgenic strain expressing a GFP-tagged R970A mutated version of CLS-2 (CLS-2$^{R970A}$). Although this transgene was expressed at similar levels to the wild-type version (*Figure 5—figure supplement 1D*), the R970A mutated protein did not localize at kinetochores, which confirmed the importance of this residue for the CLS-2 CTD function (*Figure 2H*). We next performed a yeast two-hybrid-based assay between the CLS-2 CTD wild-type or R970A and the HCP-1 CBD, which confirmed that both CLS-2$^{FL}$ and the CLS-2 CTD only, but not the R970A mutated version of the CTD, directly interact with CLS-2 and likely mediate CLS-2 kinetochore targeting (*Figure 2I*, *Figure 2—figure supplement 1G*). Overall, our results show that

BHC module assembly in meiosis and mitosis requires the BUB-1 kinase domain and the HCP-1 KTD, and involves HCP-1 CBD binding to the CLS-2 CTD through a conserved C-terminal VRK motif.

## Kinetochore and ring domain pools of the BHC module act redundantly in spindle assembly and chromosome segregation in oocytes (Figure 3)

We next tested the phenotypic effect of delocalizing the BHC module from kinetochores in meiosis by preventing its recruitment in KNL-1$^{\Delta 85\text{-}505}$ mutant oocytes. For this we performed live imaging to monitor spindle assembly and chromosome segregation in a strain expressing H2B::mCherry and GFP::β-Tubulin to label chromosomes and microtubules respectively. Depletion of endogenous KNL-1 in KNL-1$^{\Delta 85\text{-}505}$ mutant oocytes did not prevent assembly of bipolar spindles, but the spindles were smaller and displayed a reduced microtubule density (*Figure 3A and B*), demonstrating that BHC module kinetochore targeting is essential for normal spindle assembly. These shorter KNL-1$^{\Delta 85\text{-}505}$ mutant spindles were however capable of efficient chromosome segregation, unlike spindles assembled in the complete absence of KNL-1 (*Figure 3A–B*, *Video 1*, *Video 2*, *Figure 3—figure supplement 1*). The KNL-1$^{\Delta 85\text{-}505}$ mutant spindles also contrasted with oocytes fully depleted of BUB-1, which displayed severe spindle abnormalities and chromosome segregation defects (*Figure 3A*, *Video 1*, *Video 2*, *Figure 3—figure supplement 1A*). These stronger phenotypes following full BUB-1 depletion correlated with apparent delayed cell cycle progression. However, this delay, and associated spindle and chromosome segregation phenotypes, are not caused by activation of the Spindle Assembly Checkpoint (SAC), as BUB-1 itself is an essential component of the SAC and is therefore required for its functionality. Instead, the cell cycle progression delay observed in absence of BUB-1 is likely caused by its non-SAC function in promoting anaphase onset (*Kim et al., 2015*; *Kim et al., 2017*). Indeed, co-depletion of BUB-1 with another SAC component MDF-2 (ortholog of Mad2) led to an identical delay and similar defects as BUB-1 depletion alone (*Figure 3—figure supplement 1B and C*). In control oocytes with endogenous wild-type KNL-1, BHC module components localized at kinetochores and at ring domains. In KNL-1$^{\Delta 85\text{-}505}$ mutant oocytes, the BHC module still targeted to ring domains downstream of GEI-17-dependent BUB-1 sumoylation (*Figure 1D–H*). Yet in the presence of endogenous wildtype KNL-1, GEI-17-depleted oocytes formed normal bipolar spindles and chromosome segregation was accurate (*Figure 3A*, *Video 2*). We thus hypothesized that the kinetochore and ring domain pools of BHC module could act redundantly for spindle assembly and chromosome segregation in oocytes.

To determine if the mild spindle phenotype observed in KNL-1$^{\Delta 85\text{-}505}$ mutant oocytes after KNL-1 depletion is due to the persistence of the ring domain BHC pool, we depleted GEI-17 in these oocytes. Consistent with our hypothesis, the mild spindle phenotypes observed in the absence of BHC module localization at kinetochores was strongly aggravated upon simultaneous BHC delocalization at ring domains after simultaneous depletion of KNL-1 and GEI-17 in KNL-1$^{\Delta 85\text{-}505}$ mutant oocytes (*Figure 3A–B*, *Video 2*). Therefore, functions of kinetochore and ring domain pools of the BHC module are partially redundant in the control of spindle assembly and chromosome segregation. This finding also implies that CLS-2 is primarily acting at the chromosomes and not as a diffuse pool of protein along the spindle.

## BHC module integrity is essential for spindle assembly and chromosome segregation (Figure 4)

During mitosis in *C. elegans*, HCP-1/2 have been proposed to primarily act as CLS-2 kinetochore-targeting proteins (*Cheeseman et al., 2005*). We tested if this was also the case during meiosis by analyzing the effect of compromising BHC module integrity in oocytes. For this, we analyzed spindle assembly and chromosome segregation in oocytes expressing the BUB-1$^{\Delta KD}$ mutant in absence of endogenous BUB-1, and the HCP-1$^{\Delta KTD}$, or HCP-1$^{\Delta CBD}$ mutants in the *hcp-2Δ* strain following depletion of endogenous HCP-1. The three mutants should prevent BHC module assembly via different means. BUB-1$^{\Delta KD}$ blocks HCP-1/2 and CLS-2 recruitment to the kinetochores and ring domains, but does not disrupt HCP-1/2 and CLS-2 binding. HCP-1$^{\Delta KTD}$ binds CLS-2 and does not disrupt KNL-1 and BUB-1 binding, but does not localize to kinetochores. HCP-1$^{\Delta CBD}$ localizes to kinetochores, but blocks CLS-2 kinetochore recruitment (*Figure 4A*). Spindle assembly was severely disrupted in all mutants with an apolar spindle phenotype and reduced microtubule density. Meiotic chromosome segregation occurred, although lagging chromosomes during anaphase and retracted polar bodies during

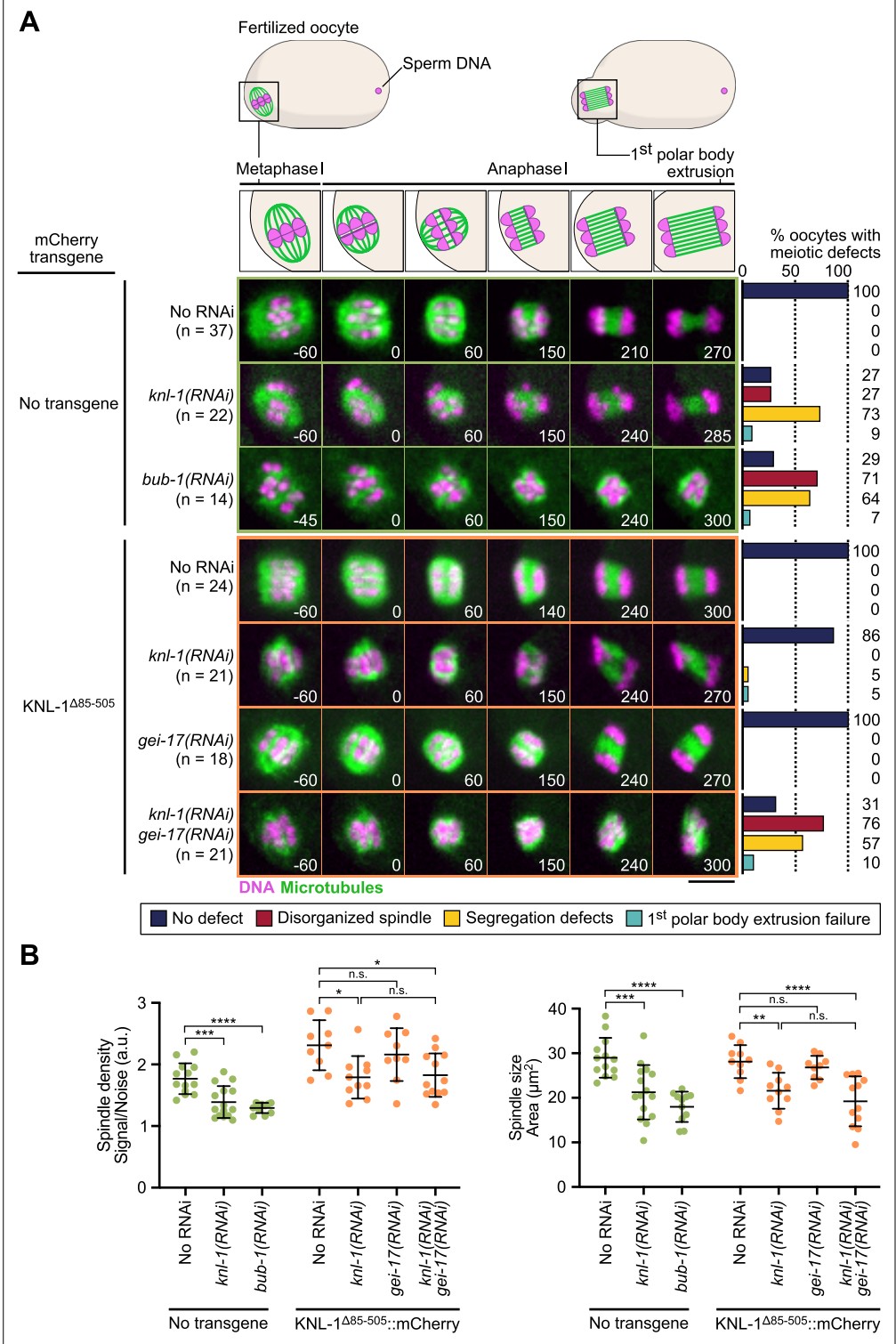

**Figure 3.** BHC module has both kinetochore-dependent and -independent functions in spindle assembly and chromosome segregation in oocytes. (**A**) Schematic of the meiotic spindle during meiosis I division (top) and stills from live imaging of meiosis I in indicated conditions (bottom). Microtubules (GFP::TBA-2$^{\alpha\text{-tubulin}}$) in green, chromosomes (mCherry::HIS-11$^{H2B}$) in magenta. Time in seconds relative to anaphase I onset. Scale bar 5 µm. Graphs on the right show quantifications of meiotic defects. (**B**) Plots of spindle density (corrected GFP intensity, left) and spindle area (right) 45 seconds before anaphase I onset. Tests, One-way ANOVA multiple comparisons

*Figure 3 continued on next page*

*Figure 3 continued*

alpha = 0.05, *p<0.05, **p<0.01, ***p<0,001, ****p<0,0001, *n.s.* not significant. Error bars, Mean and standard deviation.

The online version of this article includes the following source data and figure supplement(s) for figure 3:

**Source data 1.** Panel A source data.

**Source data 2.** Panel B source data.

**Figure supplement 1.** SAC activation is not responsible for defects observed upon BHC module perturbation.

**Figure supplement 1—source data 1.** Panel A source data.

**Figure supplement 1—source data 2.** Panel B-C source data.

telophase were evident (*Figure 4B–C*, *Video 3*, *Figure 4—figure supplement 1A*). Mitotic chromosome segregation was also strongly perturbed in the HCP-1$^{\Delta CBD}$ or HCP-1$^{\Delta KTD}$ mutants in absence of endogenous HCP-1/2 proteins, with evident sister chromatid co-segregation (*Figure 4—figure supplement 1B*). Thus, expression of BUB-1$^{\Delta KD}$, HCP-1$^{\Delta CBD}$, or HCP-1$^{\Delta KTD}$ following depletion and/or deletion of BUB-1 and HCP-1/2 recapitulated the *bub-1(RNAi)* and *hcp-1/2(RNAi)* phenotypes, with disorganized spindles and chromosome segregation defects. As in the full BUB-1 depletion, expression of BUB-1$^{\Delta KD}$ in absence of endogenous BUB-1 (or after depletion of HCP-1 in the *hcp-2Δ* mutant) also led to delayed cell cycle progression. However, this delay, and associated spindle and chromosome segregation phenotypes, were again not due to SAC activation as they could not be rescued by co-depleting the SAC component MDF-2 (*Figure 3—figure supplement 1B, C*). Overall, these results suggest that the primary function of BUB-1 and HCP-1/2 occurs in the context of the BHC module.

To further confirm that integrity of the BHC module is essential for its function, and that targeting CLS-2 to kinetochores is not sufficient for normal spindle assembly and chromosome segregation, we used protein engineering to allow CLS-2 kinetochore targeting in the absence of HCP-1/2. We engineered a fusion protein between GFP-tagged CLS-2 and a fragment of HCP-1 containing the KTD (CLS-2::GFP::HCP-1$^{1154-1386}$, *Figure 4D*), which we introduced in the *hcp-2Δ* strain and examined oocytes with and without *hcp-1(RNAi)*. We first verified that CLS-2 fused to GFP and to HCP-1$^{1154-1386}$ localized to kinetochores and to ring domains (*Figure 4E*), compared to GFP-tagged wild-type CLS2. Next to ensure that CLS-2::GFP::HCP-1$^{1154-1386}$ was functional, we checked that it could sustain embryonic development and viability in absence of endogenous CLS-2 (*Figure 4F*). Accordingly, CLS-2::GFP::HCP-1$^{1154-1386}$ also supported normal meiotic and mitotic spindle assembly and chromosome segregation upon *cls-2(RNAi)* in oocytes (*Figure 4G*, *Video 4*). Therefore, fusing CLS-2 to GFP and to HCP-1$^{1154-1386}$ did not prevent its correct localization nor functionality. CLS-2::GFP::HCP-1$^{1154-1386}$ also localized normally in absence of HCP-1 (*Figure 4E*), which suggests that the HCP-1 KTD is both necessary and sufficient for localization. However, spindles and chromosome segregation in oocytes and zygotes were severely affected upon *hcp-1(RNAi)* (*Figure 4G*).

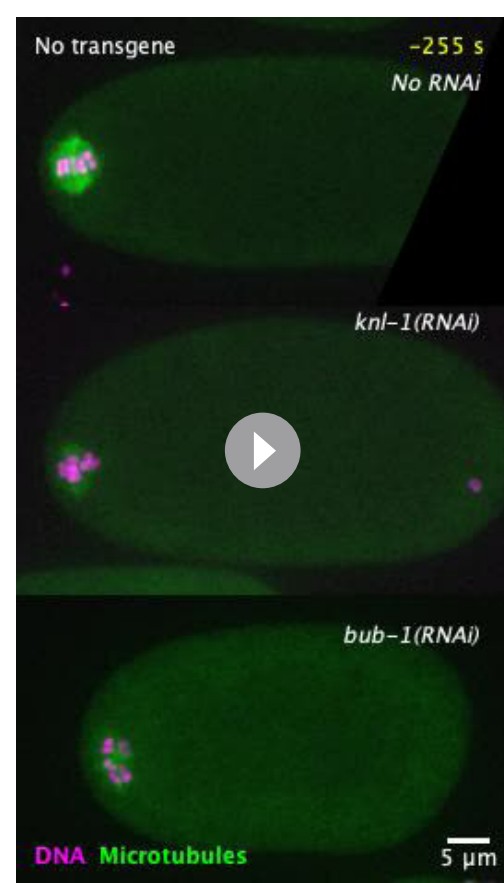

**Video 1.** Live imaging of meiosis I in indicated conditions. Microtubules (GFP::TBA-2$^{\alpha\text{-tubulin}}$) in green, DNA (mCherry::HIS-11$^{H2B}$) in magenta. Time in seconds relative to anaphase I onset. Scale bar 5 μm.
https://elifesciences.org/articles/82579/figures#video1

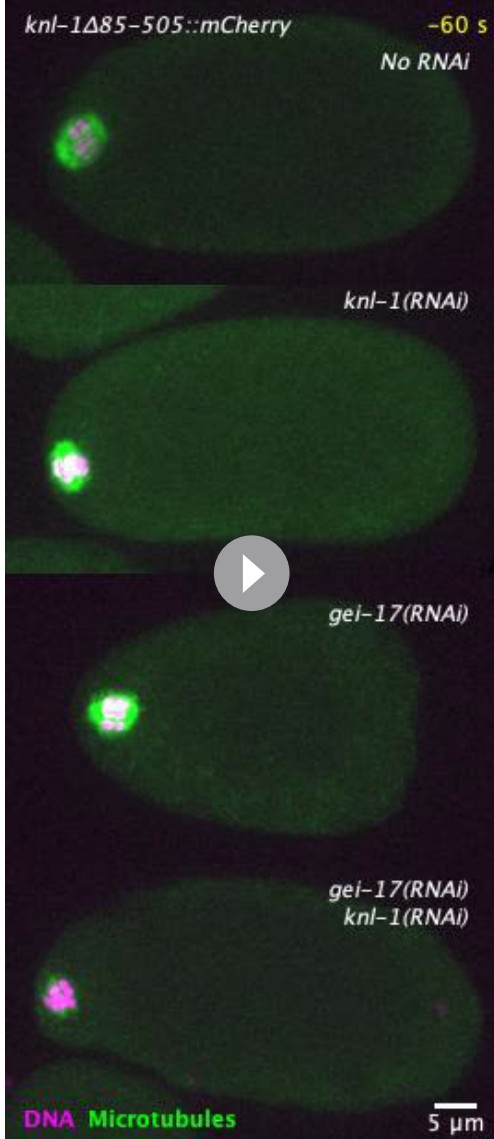

**Video 2.** Live imaging of meiosis I in *knl-1Δ85–505::mCherry* transgenic worms in indicated conditions. Microtubules (GFP::TBA-2$^{\alpha\text{-tubulin}}$) in green, DNA (mCherry::HIS-11$^{H2B}$) in magenta. Time in seconds relative to anaphase I onset. Scale bar 5 µm. https://elifesciences.org/articles/82579/figures#video2

Embryonic viability was also low in this condition (*Figure 4F*), suggesting that HCP-1/2 do not act only as targeting proteins for CLS-2 and likely have other roles in regulating microtubule dynamics in oocytes. We importantly obtained similar results when the GFP::HCP-1$^{1154-1386}$ was fused to the CLS-2$^{\Delta CTD}$ transgene, which cannot interact with endogenous HCP-1/2 (*Figure 4—figure supplement 1C–E*, *Video 5*). Overall, these results show that BHC module integrity and proper localization is essential for meiotic spindle assembly and chromosome segregation in oocytes.

## CLS-2 function does not require TOGL3 (Figure 5)

We next focused on the functional domains of CLS-2. In most species, CLASP proteins are comprised of two to three ordered TOGL (Tumor Overexpressed Gene Like 1, 2, and 3) domains (*Al-Bassam and Chang, 2011*). In human CLASP2, these domains have specific functions. TOGL2 is essential for microtubule catastrophe suppression, whereas TOGL3 enhances rescue (*Lawrence et al., 2018*; *Aher et al., 2018*; *Girão et al., 2020*). The C-terminal domain (CTD) of CLASPs, responsible for kinetochore targeting, can inhibit these functions in human CLASP2, while the TOGL1 is required to release this auto-inhibition (*Aher et al., 2018*). Additionally, a conserved Serine/Arginine-rich (S/R-rich) region, important for microtubule lattice binding through electrostatic interactions, separates TOGL2 and 3 (*Al-Bassam et al., 2010*; *Patel et al., 2012*). Sequence analysis showed that CLS-2 contains only two TOGL domains (TOGL2 and 3) separated by an S/R-rich region, and a CTD separated from TOGL3 by a linker region predicted to be largely unfolded and a small also unfolded C-terminal tail (*Figure 5—figure supplement 1A–C*).

We generated a series of transgenic strains expressing RNAi-resistant GFP-fused truncations in CLS-2 or point mutations in conserved and functionally important residues (*Figure 5A*, *Figure 5—figure supplement 2*), and we quantified embryonic viability upon depletion of endogenous CLS-2. All transgenes were expressed at comparable levels (*Figure 5—figure supplement 1D*), but only five mutants were unable to sustain embryonic viability in absence of endogenous CLS-2. These corresponded to deletion or point mutations within TOGL2 (CLS-2$^{\Delta TOGL2}$, CLS-2$^{TOGL2-WKR/A}$), the S/R-rich region (CLS-2$^{\Delta SR-rich}$), and the CTD (CLS-2$^{\Delta CTD}$, CLS-2$^{R970A}$) (*Figure 5B*). In contrast, deleting TOGL3, or the linker region between TOGL3 and the CTD, or the C-terminal tail had no significant effect on embryonic viability upon depletion of endogenous CLS-2 (*Figure 5B*). Sequence analysis revealed that TOGL3 lacks several conserved amino-acids (present in TOGL2 and mutated in the CLS-2$^{TOGL2-WKR/A}$ mutant, *Figure 5—figure supplement 2*) corresponding to residues that normally contribute to microtubule binding in human CLASPs and to tubulin binding in XMAP215 family proteins (*Leano et al., 2013*; *Funk et al., 2014*). This evolutionary sequence divergence could potentially account for

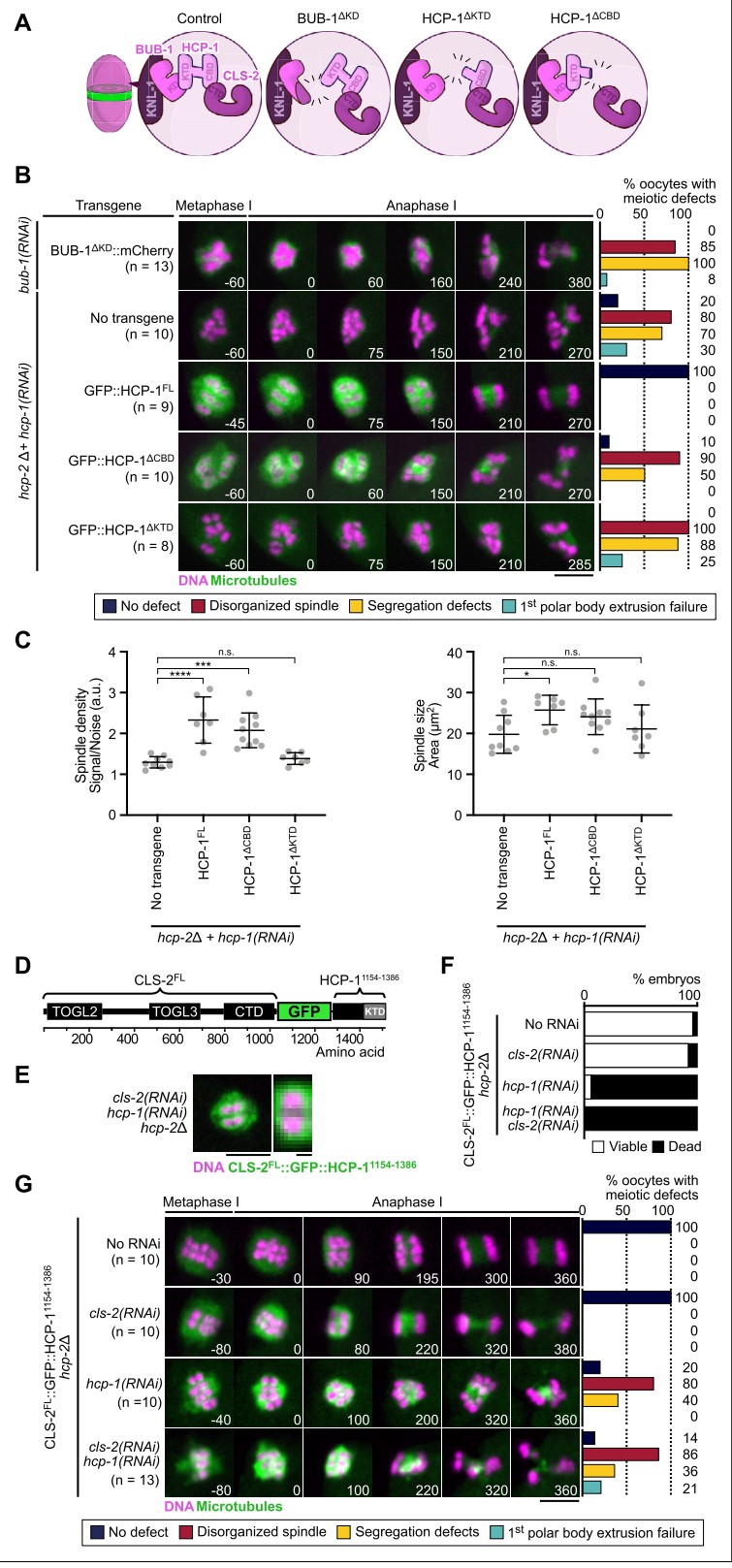

**Figure 4.** BHC module integrity is essential for spindle assembly and accurate chromosome segregation.
(**A**) Schematic of BHC-module integrity mutants. (**B**) Stills from live imaging of meiosis I in indicated conditions.
Microtubules (GFP::TBA-2$^{\alpha\text{-tubulin}}$ or GFP::TBB-2$^{\beta\text{-tubulin}}$) in green, chromosomes (mCherry::HIS-11$^{H2B}$) in magenta.
Time in seconds relative to anaphase I onset. Graphs indicate quantifications as referred to in color key. (**C**) Plots of

*Figure 4 continued on next page*

*Figure 4 continued*

spindle density (corrected GFP intensity, left) and spindle area (right) 45 s before anaphase I onset. Kruskal-Wallis multiple comparisons, alpha = 0.05, *p<0.05, ***p<0.001, ****p<0.0001, *n.s.* not significant. Error bars, Mean with standard deviation. (D) Schematic of the CLS-2$^{FL}$::GFP::HCP-1$^{1154-1386}$ protein fusion. The HCP-1$^{1154-1386}$ fragment contains the KTD. (E) Localization of CLS-2$^{FL}$::GFP::HCP-1$^{1154-1386}$ (green) in metaphase I, and magnification of a single meiosis I chromosome. DNA (mCherry::HIS-11$^{H2B}$) in magenta (n=13). (F) Embryonic lethality and (G) meiotic defects rescue assays of indicated depletions by CLS-2$^{FL}$::GFP::HCP-1$^{1154-1386}$. Scale bars, full spindle 5 µm, single chromosome details 1 µm.

The online version of this article includes the following source data and figure supplement(s) for figure 4:

**Source data 1.** Panel B source data.

**Source data 2.** Panel C source data.

**Source data 3.** Panel F source data.

**Source data 4.** Panel G source data.

**Figure supplement 1.** Artificial kinetochore and ring localization of CLS-2 is not sufficient to rescue loss of BHC module integrity.

**Figure supplement 1—source data 1.** Panel A source data.

**Figure supplement 1—source data 2.** Panel E source data.

---

the apparent lack of requirement for TOGL3, as mutating the corresponding residues in TOGL2 abrogated CLS-2 functionality (*Figure 5B and D*). As indicated previously, only deleting or mutating the CTD prevented chromosomal localization of the corresponding GFP-tagged transgene (*Figure 5C*), suggesting that the other three transgenes carried loss-of-function mutations.

We confirmed a loss-of-function phenotype by analyzing meiotic spindle assembly and chromosome segregation in oocytes lacking endogenous CLS-2. Mutating or deleting TOGL2, the S/R-rich region or the CTD led to phenotypes comparable to the full loss of function of CLS-2 with severely disorganized spindles, inaccurate chromosome segregation and polar body extrusion failures (*Figure 5D*, *Video 6*). While mutations in the TOGL2 or S/R-rich region usually led to complete abrogation of chromosome segregation (70% and 92% oocytes did not attempt chromosome segregation, respectively), and to polar body extrusion failure (80% and 100% oocytes, respectively), oocytes lacking (CLS-2$^{\Delta CTD}$) or mutated in the CLS-2 CTD (CLS-2$^{R970A}$) frequently displayed attempted chromosome segregation (only 18% and 50% oocytes did not attempt chromosome segregation, respectively) and often successfully extruded the first polar body (36% and 50% polar body extrusion failure, respectively). Chromosome segregation defects were also observed in mitosis in the presence of the corresponding transgenes upon depletion of endogenous CLS-2 (*Figure 5—figure supplement 1E*). Thus, as in human CLASP2, CLS-2 functions via a single TOGL domain (TOGL2), and also requires the CTD for HCP-1 binding and the S/R-rich region for potential microtubule lattice-interaction.

## BHC module components synergistically stabilize microtubules in vitro (Figure 6)

To investigate the effect of BHC module components on microtubule dynamics, we purified full-length BUB-1, HCP-1, and GFP-tagged CLS-2 from insect cells and analyzed their activity using an in vitro microtubule-based assay (*Figure 6—figure supplement 1A*). Microtubule growth from GMPCPP-stabilized seeds was monitored by TIRF (Total Internal Reflection Fluorescence) microscopy at 23 °C (*Aumeier et al., 2016*; *Colin et al., 2018*; *Figure 6A*). Consistent with BUB-1 not being a microtubule-associated protein (MAP), analysis of microtubule dynamics showed that 100 nM BUB-1 did not display any strong effect on the microtubule growth rate, on catastrophe and rescue frequencies, nor on the time spent in pause by microtubules (*Figure 6B–F*, *Video 7*). BUB-1 on its own also did not interact with microtubules in a pelleting assay (*Figure 6—figure supplement 1B*), nor show any microtubule-bundling activity (*Figure 6G*). In contrast, both HCP-1 and CLS-2-GFP independently displayed strong microtubule-bundling activity (*Figure 6G*). Consistent with a previous study showing that human CENP-F can stimulate microtubule polymerization in vitro *Feng et al., 2006*, 100 nM HCP-1 had a mild but significant promoting effect on the microtubule growth rate (1.4-fold increase). This was interestingly abrogated by addition of 100 nM BUB-1, but not by a combination of BUB-1 and CLS-2-GFP (100 nM each; *Figure 6B–F*, *Video 7*). In contrast, 100 nM CLS-2-GFP

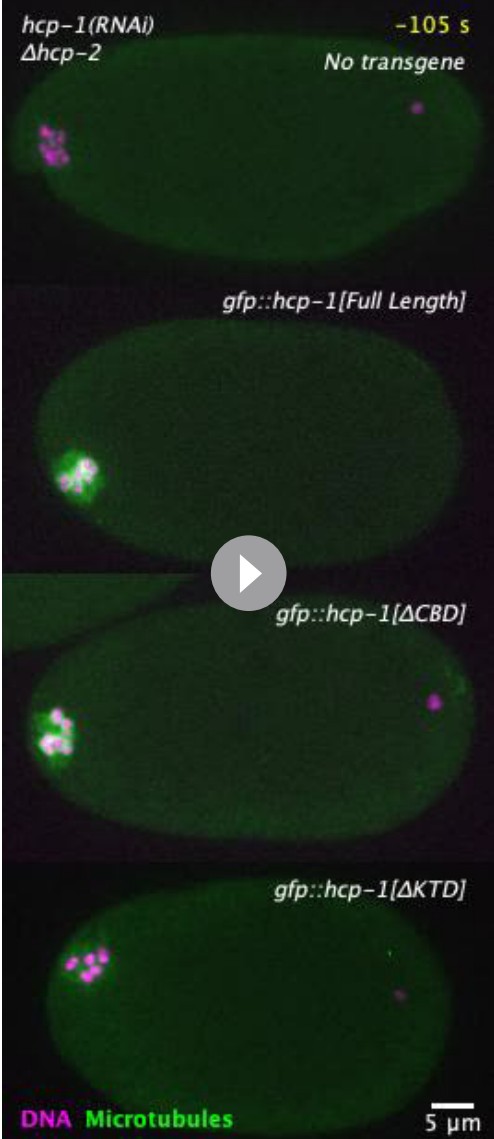

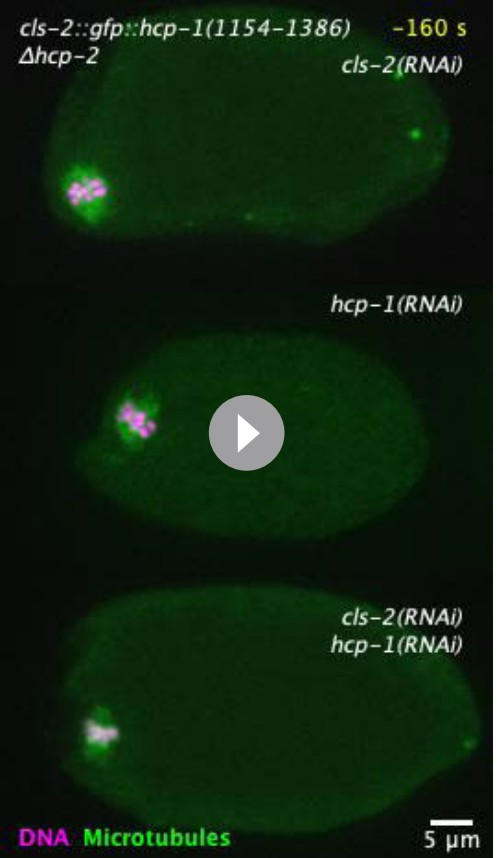

**Video 3.** Live imaging of meiosis I in indicated transgenic *hcp-2*-mutant worms upon depletion of endogenous *hcp-1*. Microtubules (GFP::TBA-2$^{\alpha\text{-tubulin}}$) in green, DNA (mCherry::HIS-11$^{H2B}$) in magenta. Time in seconds relative to anaphase I onset. Scale bar 5 μm.
https://elifesciences.org/articles/82579/figures#video3

**Video 4.** Live imaging of meiosis I in *hcp-2*-mutant worms expressing CLS-2::GFP::HCP-11$^{1154-1386}$ fusion protein in indicated conditions. Microtubules (GFP::TBA-2$^{\alpha\text{-tubulin}}$) in green, DNA (mCherry::HIS-11$^{H2B}$) in magenta. Time in seconds relative to anaphase I onset. Scale bar 5 μm.
https://elifesciences.org/articles/82579/figures#video4

alone had the opposite effect and inhibited the microtubule growth rate (1.1-fold reduction), which is consistent with the effect of human or *Drosophila* CLASPs on microtubule dynamics (*Yu et al., 2016*; *Moriwaki and Goshima, 2016*; *Aher et al., 2018*). Also consistent with previous reports on CLASPs, we found that CLS-2-GFP had significant catastrophe-suppressing, and rescue- and pause-promoting activities (*Figure 6B–F*, *Video 7*; *Al-Bassam et al., 2010*; *Lawrence et al., 2018*; *Aher et al., 2018*). Furthermore, and like human CLASPs, CLS-2 is unlikely to function as a microtubule polymerase, as neither of the two TOGL domains interacted with free tubulin (*Figure 6—figure supplement 1C*; *Aher et al., 2018*). The effect of CLS-2-GFP on catastrophe was independent of HCP-1 and/or BUB-1 (*Figure 6C–F*). This is in contrast to its effect on rescue promotion, which was increased by addition of HCP-1 alone (twofold increase), but not by a combination of HCP-1 and BUB-1. The most surprising effect of reconstituting the full BHC module (BUB-1, HCP-1, and CLS-2-GFP together) in vitro was the strong increase in the pause-promoting activity (21-fold increase in the time spent in pause) compared to all other conditions tested (*Figure 6F*, *Video 7*). Thus, reconstituting the BHC module in vitro displays both additive (promotion of the growth rate and rescue frequency by HCP-1 and CLS-2 respectively, and inhibition of catastrophe by CLS-2) and synergistic (promotion of pause) effects on microtubule dynamics

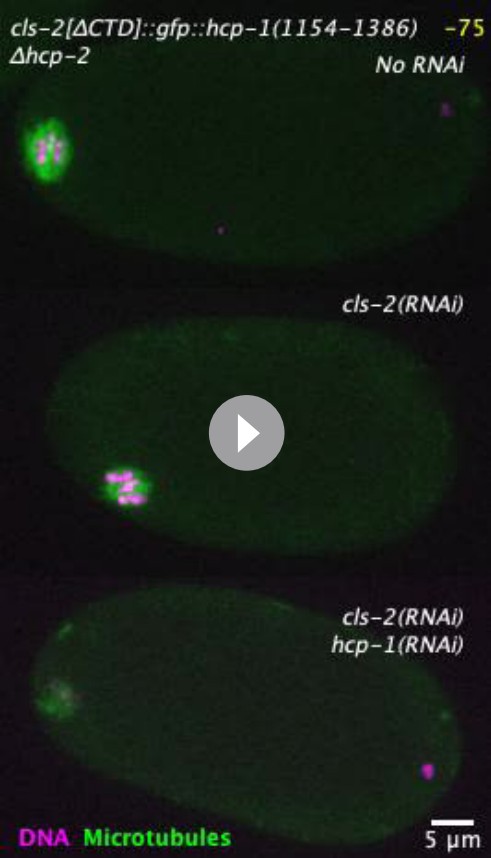

**Video 5.** Live imaging of meiosis I in *hcp-2*-mutant worms expressing CLS-2$^{\Delta CTD}$::GFP:: HCP-1$^{1154-1386}$ fusion protein in indicated conditions. Microtubules (GFP::TBA-2$^{\alpha-tubulin}$) in green, DNA (mCherry::HIS-11$^{H2B}$) in magenta. Time in seconds relative to anaphase I onset. Scale bar 5 µm.

https://elifesciences.org/articles/82579/figures#video5

compared to individual components, which leads to microtubule stabilization. This also indicates that BUB-1 and HCP-1 can modulate the effect of CLS-2 on microtubule dynamics. To determine if the additive and synergistic effects of BHC module components on microtubule dynamics depended on the formation of an intact BHC module, we analyzed microtubule dynamics in the presence of the CLS-2$^{R970A}$-GFP mutant, incapable of binding to HCP-1. Importantly, this mutant alone had identical effects on all microtubule dynamics parameters tested, as compared to wild-type CLS-2-GFP, suggesting that CLS-2$^{R970A}$-GFP was functional. However, it did not display the same increase of microtubule pausing when combined with HCP-1 and BUB-1, showing that this synergistic effect specifically required formation of an intact BHC module. Overall, our in vitro results are consistent with our in vivo data and suggest BUB-1 and HCP-1/2 in the context of an intact BHC module are important contributors in the regulation of microtubule dynamics.

## Discussion

Our results highlight the importance and molecular mechanisms by which a kinetochore module, the BHC module, comprising the kinase BUB-1, the two CENP-F orthologs HCP-1/2 and the CLASP family member CLS-2, regulate microtubule dynamics in *C. elegans* oocytes and zygotes for efficient chromosome segregation.

### BHC module assembly and kinetochore targeting

We found that assembly of the BHC module requires the kinase domain of BUB-1 and the KTD of HCP-1, although we were unable to show a direct interaction between these domains using a yeast-two-hybrid assay. This may be a false negative due to proteins not being properly expressed/folded in yeast. Supporting this view, in humans, the kinase domain of Bub1 interacts directly with a C-terminal domain of CENP-F, and this interaction is necessary for CENP-F kinetochore targeting, which is reminiscent of our findings in *C. elegans* oocytes and zygotes (*Ciossani et al., 2018*). Also consistent with previous findings in human cells and with the fact that a kinase dead mutant of BUB-1 can sustain embryonic viability in *C. elegans*, we show that the kinase domain, but not kinase activity of BUB-1, is required for the localization of the CENP-F-like protein HCP-1 to kinetochores (*Moyle et al., 2014*). Finally, we found that CLS-2 kinetochore recruitment involves a direct interaction between the HCP-1 CBD and CLS-2 CTD. In humans, CLASP kinetochore localization was also shown to depend on its CTD (*Maiato et al., 2003*). However, in contrast to our present results, CENP-E, but not CENP-F, is responsible for kinetochore targeting of CLASP (*Maffini et al., 2009*). Interestingly, CENP-E and CENP-F have been speculated to be distantly related paralogs (*Ciossani et al., 2018*). The fact that in *C. elegans*, which lack a CENP-E ortholog, the CENP-F-like protein HCP-1 is responsible for CLASP kinetochore localization supports this hypothesis.

A surprising finding was that kinetochore targeting of the BHC module could occur independently of BUB-3 in *C. elegans* oocytes, which is in contrast with previous findings in *S. cerevisiae* and human cells (*Primorac et al., 2013*; *Vleugel et al., 2013*). Although this BUB-3-independent kinetochore

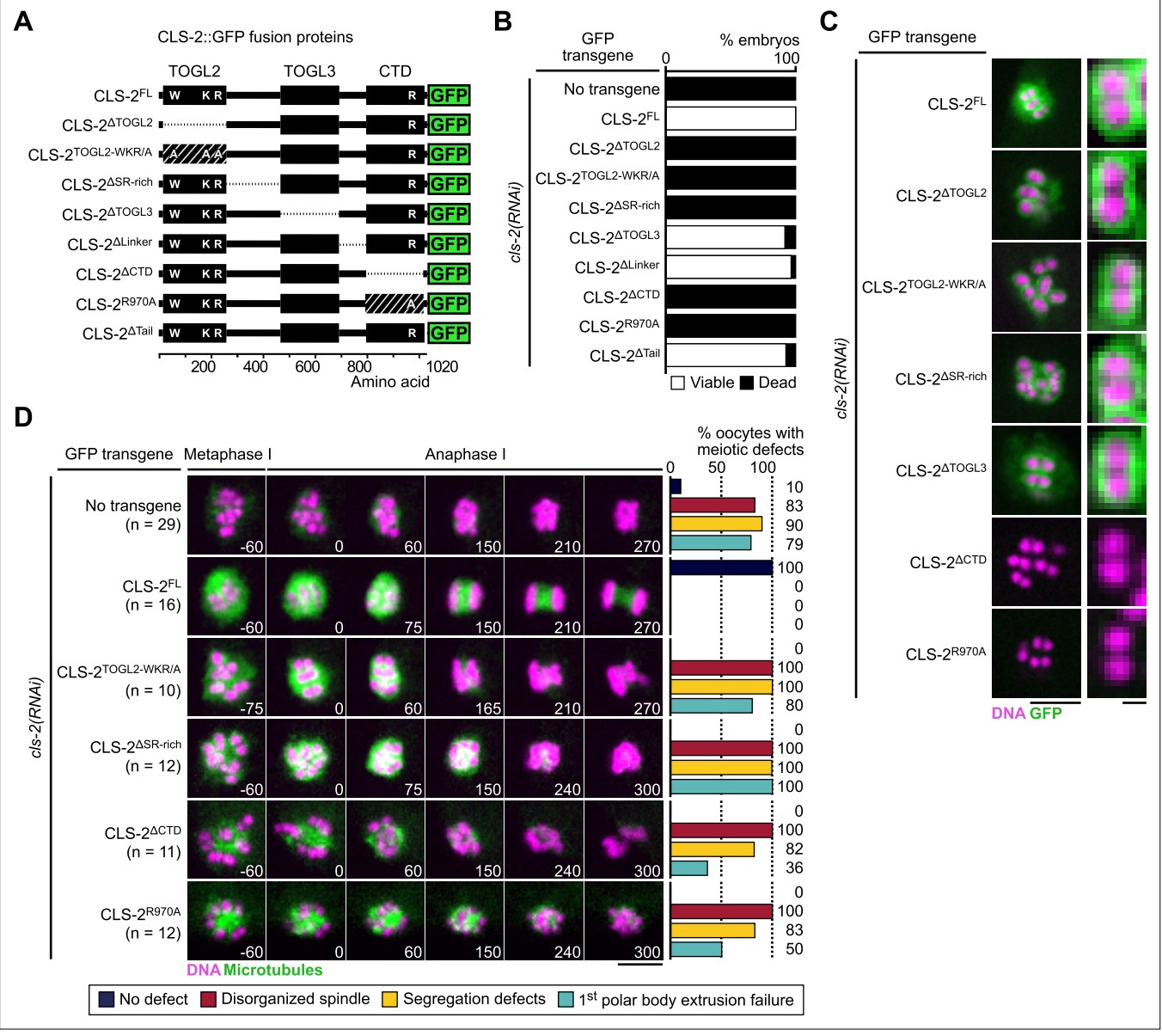

**Figure 5.** CLS-2 function does not require TOGL3. (**A**) Schematic of CLS-2::GFP truncated fusions. (**B**) Embryonic viability assay upon depletion of endogenous *cls-2* in the presence of indicated transgene. (**C**) Localization of CLS-2::GFP truncations (green) during metaphase I in indicated conditions. DNA (mCherry::HIS-11^H2B) in magenta (n≥9). (**D**) Stills from live imaging of meiosis I. Microtubules (GFP::TBA-2^α-tubulin) in green, chromosomes (mCherry::HIS-11^H2B) in magenta. Time in seconds relative to anaphase I onset. Graphs indicate quantifications of meiotic defects. Scale bars, full spindles 5 μm, single chromosome details 1 μm.

The online version of this article includes the following source data and figure supplement(s) for figure 5:

**Source data 1.** Panel B source data.

**Source data 2.** Panel D source data.

**Figure supplement 1.** Protein domains essential for CLS-2 function.

**Figure supplement 1—source data 1.** Panel D source data.

**Figure supplement 2.** Protein sequence alignment of indicated eukaryotic TOGL domains.

localization of BUB-1 could be meiosis-specific, we suspect that it is rather a *C. elegans* adaptation. Indeed, some BUB-1 is also visible at kinetochores in *C. elegans* zygotes during mitosis in the *bub-3Δ* mutant (**Kim et al., 2015**). How then is BUB-1 targeted to kinetochores in *C. elegans*? We envision two plausible scenarios. First, although Bub3, in yeasts and mammals, is the primary determinant

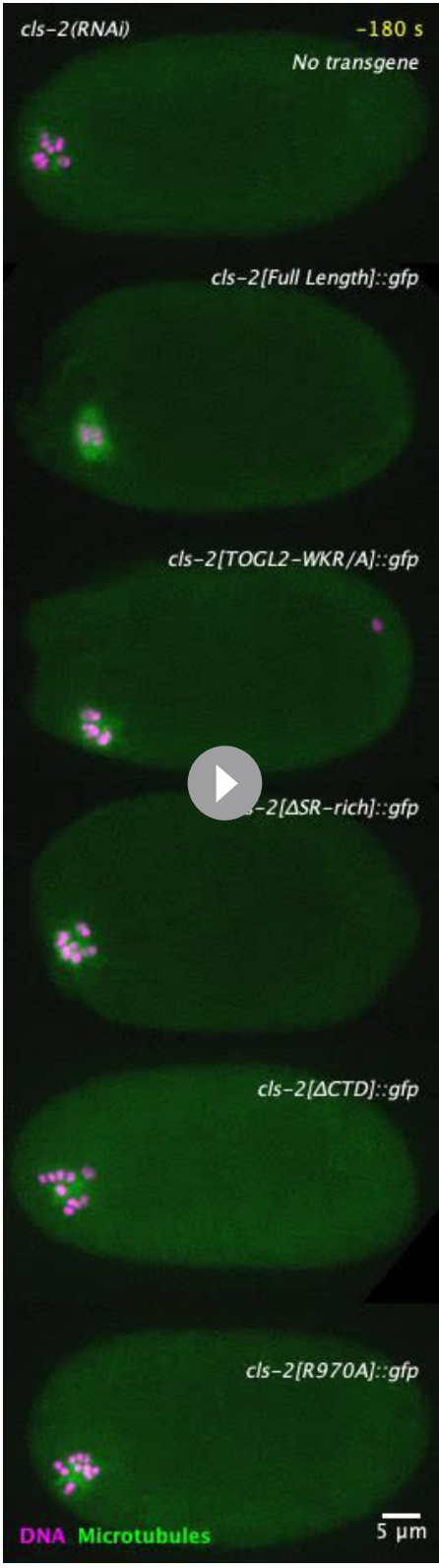

**Video 6.** Live imaging of meiosis I in indicated transgenic worms upon depletion of endogenous *cls-2*. Microtubules (GFP::TBA-2$^{α\text{-tubulin}}$) in green, DNA (mCherry::HIS-11$^{H2B}$) in magenta. Time in seconds relative to anaphase I onset. Scale bar 5 µm.

https://elifesciences.org/articles/82579/figures#video6

of Knl1 MELT binding, structural analysis demonstrated that direct contacts, which normally reinforce Bub3 binding, exist between a unique loop in Bub1 and the Knl1 MELT repeats (*Overlack et al., 2015*). In *C. elegans*, these contacts could be sufficient for BUB-1 recruitment to kinetochores in absence of BUB-3. Second, prior to the identification of the primary role of Knl1 MELT repeats for Bub1/Bub3 kinetochore targeting, a prevalent model implicated interaction between a KI (Lys-Ile) motif in Knl1, which lies in the MELT repeat region of the protein, and the conserved TPR domain of Bub1 (*Kiyomitsu et al., 2007*; *Kiyomitsu et al., 2011*). Although, the KI motif was later shown to only act as a MELT-enhancing motif for Bub3-binding, and no clear equivalent motif could be identified in *C. elegans* KNL-1, it is possible that a sequence-divergent, but functionally equivalent, motif exists in KNL-1 that would be sufficient for recruiting BUB-1 to kinetochores (*Krenn et al., 2014*). Furthermore, the fact that in *C. elegans* a *bub-3Δ* mutant is viable and fertile implies that it does not display any of the severe meiotic and mitotic phenotypes associated with loss-of-function of the BHC module (*Kim et al., 2015*). This in turn indicates that the reduced BHC pools, recruited at kinetochores and ring domains independently of BUB-3, are functional. Further functional analysis of BUB-1 kinetochore targeting in this system will be required to elucidate the BUB-3-independent mechanism of BHC module localization.

## BHC module function in meiotic spindle assembly and chromosome segregation

Our findings demonstrate that integrity of the BHC module is essential in vivo in *C. elegans* oocytes for functional meiotic spindle assembly. Specifically, our quantitative analysis of meiotic spindle assembly and function, demonstrated that depletion of BHC module components individually, or perturbation of BHC module assembly, gave rise to disorganized spindles with reduced microtubule densities. Although mammalian CLASPs have also been involved in proper spindle assembly and mitotic fidelity, this contrasts with previous work in mammals, which demonstrated that CENP-F is largely dispensable for spindle assembly and chromosome segregation (*McKinley and Cheeseman, 2017*; *Raaijmakers et al., 2018*; *Logarinho et al., 2012*; *Haley et al., 2019*). While in mammals CENP-E, and not CENP-F, is responsible for localizing CLASPs at kinetochores, a critical function of *C.*

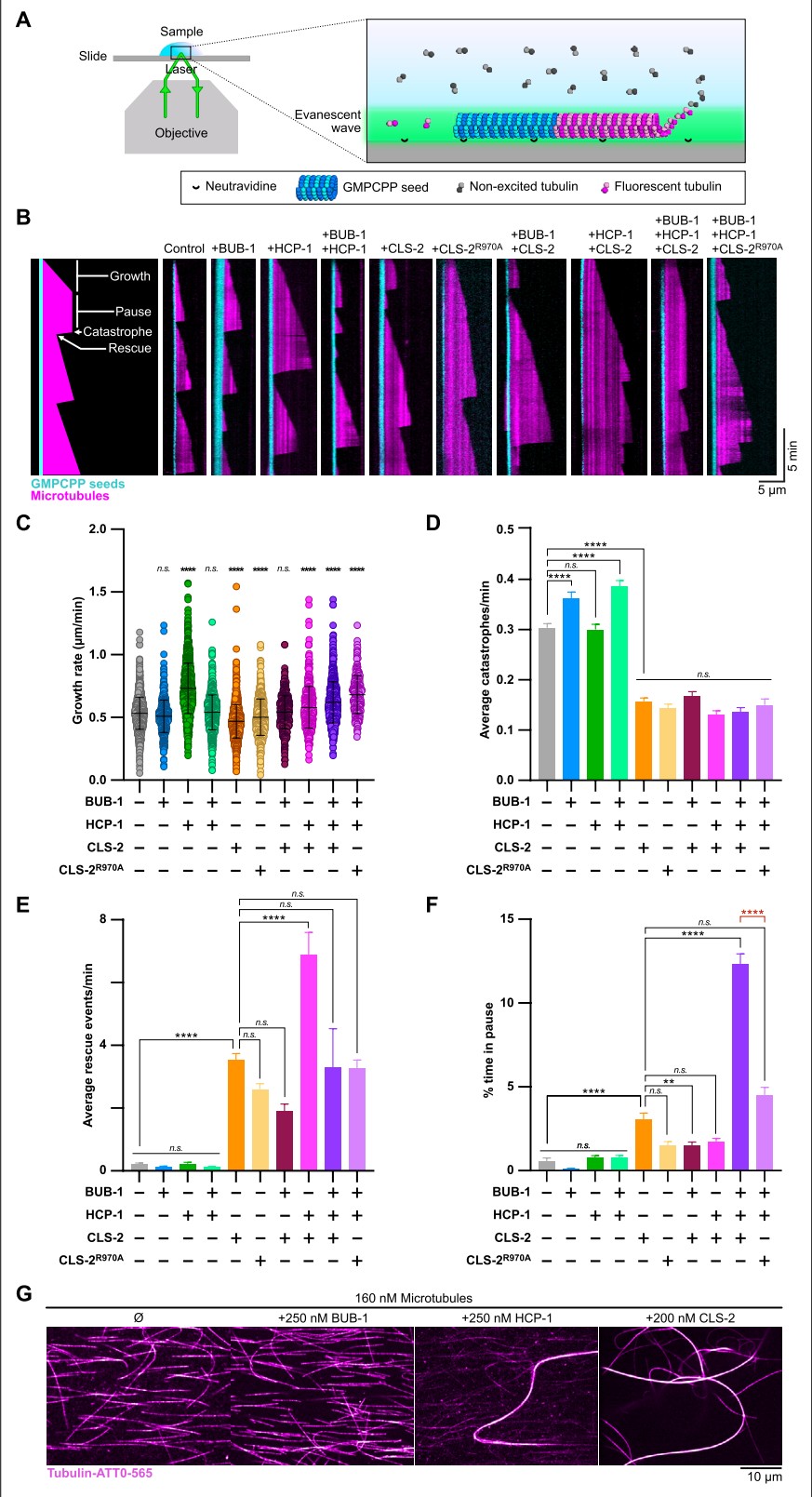

**Figure 6.** BHC module components synergistically stabilize microtubules in vitro. (**A**) Schematic of the TIRF microscopy-based microtubule assay. Labeled tubulin (ATTO-565, magenta) fluoresces only when close to the surface of the coverslip. Microtubules polymerize from biotinylated GMPCPP seeds (tubulin-ATTO-488, cyan) bound to a Neutravidin-coated glass coverslip. (**B**) Representative kymographs of microtubules (magenta)

*Figure 6 continued on next page*

*Figure 6 continued*

growing from GMPCPP seeds (cyan) in the presence or absence of BUB-1, HCP-1, CLS-2-GFP and/or CLS-2$^{R970A}$-GFP (100 nM each). Schematics on the left highlights the different microtubule dynamics events observed. (**C–F**) Dot plot showing the quantification of growth rate (**C**), and histograms showing the average catastrophe (**D**), rescue (**F**), and pause (**D**) events per microtubule. Dunnett's multiple comparison tests, alpha = 0.01, **$p<0.01$, ****$p<0.0001$, *n.s.* not significant. Error bars, Mean and standard deviation (**C**) or standard error of the mean (**D–F**). (**G**) Microtubule bundling assay. Organization of microtubules (magenta) observed in indicated conditions.

The online version of this article includes the following source data and figure supplement(s) for figure 6:

**Source data 1.** Panel B-F source data.

**Source data 2.** B-F statistics source data.

**Figure supplement 1.** CLS-2 decorates the microtubule lattice in vitro.

**Figure supplement 1—source data 1.** Panels A-B source data.

**Figure supplement 1—source data 2.** Panel C source data.

**Figure supplement 1—source data 3.** Panel D source data.

*elegans* HCP-1/2 is to target the CLASP family member CLS-2 to kinetochores and ring domains (*Maffini et al., 2009*). Yet, our in vivo results show that localizing CLS-2 to kinetochores and rings independently of HCP-1/2 is not sufficient to rescue the spindle phenotype of HCP-1/2-depleted oocytes. This argues for HCP-1/2 playing additional essential roles in regulating microtubule dynamics and spindle function, beside simply promoting CLS-2 localization.

In human cells, CENP-F binds to microtubules through two independent MTBDs located at either terminus of the protein, both of which are required for normal tension at centromeres in human cells (*Feng et al., 2006*; *Volkov et al., 2015*; *Kanfer et al., 2017*; *Auckland et al., 2020*). Our in vitro experiments are consistent with HCP-1 also binding directly to microtubules, although the domain(s) responsible for this activity has not been identified. Human CENP-F was also shown to stimulate microtubule assembly in a bulk in vitro assay through an unknown mechanism (*Feng et al., 2006*). We interestingly show here that the *C. elegans* orthologous protein HCP-1 increases the microtubule growth rate in vitro. Whether microtubule binding and/or microtubule growth rate promotion, independent of CLS-2 targeting, are the important functions of HCP-1 explaining its direct role in meiotic spindle assembly is unclear. We interestingly found that HCP-1 could synergize the rescue promoting effect of CLS-2 in vitro (twofold increase in the rescue frequency), but that this synergy was lost upon addition of BUB-1. Importantly, this cannot be caused by a competition between BUB-1 and CLS-2, for HCP-1 binding, because the pause promoting effect was only observed when all three BHC components were added. Further work, including identification of separation-of-function mutations of HCP-1, is clearly required to establish the exact function of HCP-1 in vivo.

Our data also show that the CTD point or deletion mutant of CLS-2 (CLS-2$^{R970A}$ or CLS-2$^{\Delta CTD}$), which were unable to interact with HCP-1 or support normal meiotic spindle assembly, could largely support chromosome segregation and polar body extrusion, unlike other CLS-2 loss-of-function mutants. This suggests that CLS-2$^{R970A}$ and CLS-2$^{\Delta CTD}$ have retained some functions of CLS-2, and extends previous findings showing that CLS-2 can localize to, and promote assembly of, central spindle microtubules in absence of its upstream partner protein BUB-1, although less efficiently (*Laband et al., 2017*). Consistent with this view, our in vitro microtubule-based assays showed that, as wild-type CLS-2, the CLS-2$^{R970A}$ mutant was capable of promoting microtubule rescue, while inhibiting catastrophe. Our assays also demonstrated that, as a module, BHC components strongly enhanced microtubule pausing relative to individual components individually or in pairs. Altogether, our data suggest that meiotic spindle assembly, chromosome segregation, and polar body extrusion are distinct functions of CLS-2 requiring different levels of microtubule dynamics regulation. Meiotic spindle assembly requires an intact BHC module and the

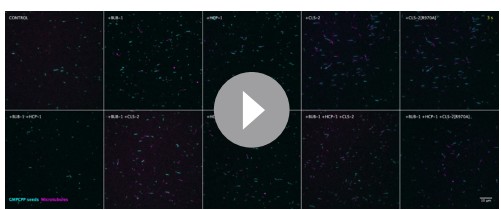

**Video 7.** TIRF microscopy-mediated live imaging of in vitro microtubule (magenta) polymerization dynamics from GMPCPP seeds (cyan) in the presence of indicated protein (100 nm each). Scale bar 10 μm.

https://elifesciences.org/articles/82579/figures#video7

strong microtubule pause promoting effect of the module, while chromosome segregation and polar body extrusion only depend on the rescue promoting and catastrophe reducing effects of CLS-2 alone.

In mammals, CLASPs contain two tandemly arranged SxIP motifs that mediate End-Binding (EB) protein interaction important for microtubule plus-end accumulation and microtubule catastrophe suppression (*Aher et al., 2018*; *Girão et al., 2020*; *Patel et al., 2012*; *Maki et al., 2015*). In *C. elegans* oocytes and embryos, CLS-2 did not substantially accumulate at microtubule plus-ends, which was consistent with the apparent lack of SxIP motif in the CLS-2 sequence. We noted that CLS-2 contained a divergent LxxPTPh motif (LPKRPTPQ) at the C-terminal end of the S/R-rich region, which could potentially mediate EB protein interaction (*Kumar et al., 2017*). However, a GFP-fused deletion of the corresponding region could support embryonic viability in absence of endogenous CLS-2, which suggests that it is not essential (*Figure 6—figure supplement 1D*). Also, in contrast to human CLASP2, which weakly but visibly accumulates at microtubule plus-ends in vitro in the absence of EB protein, GFP-tagged CLS-2 only faintly decorated the microtubule lattice, with a speckled pattern (*Figure 6—figure supplement 1E*). This weak microtubule binding was not increased in the presence of HCP-1 and BUB-1 (*Figure 6—figure supplement 1E*). Finally, in contrast to work on yeasts and *Drosophila* CLASPs, but in line with previous in vitro studies on human CLASPs, we did not detect any significant correlation between the CLS-2 speckles along the lattice and the sites of microtubule rescue or pause events (*Figure 6—figure supplement 1F*; *Al-Bassam et al., 2010*; *Moriwaki and Goshima, 2016*; *Lawrence et al., 2018*). Therefore, unlike mammalian, or yeasts and *Drosophila* CLASPs, which respectively prevent catastrophe and promote rescue by concentrating at a region located behind the outmost microtubule end, or by accumulating at speckles along the lattice, the stabilizing effect of *C. elegans* CLS-2 does not require its plus-end nor lattice speckled accumulation.

## Potential conservation of BHC function in oocytes outside of *C. elegans*

Although BHC components are mostly conserved across evolution from fly, to worm, to mammals, their function as a module, such as the one we described here in *C. elegans*, is unlikely to be identically conserved. Indeed, although CENP-F kinetochore localization depends on its direct interaction with Bub1, CENP-E and not CENP-F is responsible for CLASPs kinetochore localization in human cells. This could explain why, unlike HCP-1/2 in *C. elegans*, CENP-F is not an essential protein in vertebrates (*Pfaltzgraff et al., 2016*; *McKinley and Cheeseman, 2017*; *Raaijmakers et al., 2018*). In mammals, CENP-E interacts with the pseudokinase domain of BubR1, but does not bind to the kinase domain of Bub1 (*Ciossani et al., 2018*). Overall, we showed here that, *in C. elegans*, binding to HCP-1 and BUB-1 could modulate CLS-2 effect on microtubules. As BubR1 and Bub1 are clear paralogous proteins, and CENP-E and CENP-F have been proposed to be distantly related paralogs, it would be interesting to determine if similar synergistic regulation of microtubule dynamics exists between BubR1, CENP-E and CLASP proteins in vertebrates (*Ciossani et al., 2018*). More generally, determining if other CLASPs partner proteins can also modulate their microtubule dynamics-regulating activity will be essential to fully understand CLASPs function.

## Methods
### *C. elegans* strain maintenance

*C. elegans* strains were maintained at 20° or 23 °C under standard growth conditions on NGM plates and fed with OP50 *E. coli* (*Brenner, 1974*). The N2 Bristol strain was used as the wild-type control background *Sulston and Brenner, 1974*, unless specified otherwise in the text. Transgenic lines were obtained either by Mos1-mediated Single Copy Insertion (MoSCI) *Frøkjaer-Jensen et al., 2008*, or by crossing pre-existing strains. JDU753 was obtained by Crispr/Cas9-mediated mutagenesis of JDU244 using crJD66 (5'- cttcgacttcaatgacacga-3') and crJD67 (5'- tccgagatcaagcagctgaa-3') guide RNAs. We used 5'-gcttggagaaatgcatgagaagatggaagcatctga(T)cgtcgtgtgtcattgaagtcgaagaacaa^ccgccacgagtaa tgggactgatcgagcaagcaag-3' as repair template, where (T) corresponds to a silent mutation in one of the PAM, and '^' indicates the deleted KTD. A potential mutation of endogenous *hcp-1* was eliminated through backcrosses, while the mutated *gfp::hcp-1* MoSCI transgene was specifically selected for by PCR. The list of strains used in this study is provided in . All strains produced will be provided

upon request and/or made available through the *Caenorhabditis* Genetics Center (CGC, https://cgc.umn.edu/).

## RNA-mediated interference

Double-stranded RNAs were produced as described previously in *Edwards et al., 2018*. After DNA amplification from total N2 cDNA by PCR using the primers listed below, reactions were cleaned (PCR purification kit, Qiagen), and used as templates for T3 and T7 transcription reactions (MEGAscript,Invitrogen) for 5 hr at 37 °C. These reactions were purified (MEGAclear kit, Invitrogen), then annealed at 68 °C for 10 minutes, 37 °C for 30 min. L4-stage hermaphrodite larvae were micro-injected with 1500–2000 µg/µL of each dsRNA, and recovered at 20 °C for 36 hr before being further processed.

> Primers used to produce double-stranded RNAs targeting the indicated genes: *bub-1* (R06C7.8):
> 5′-AATTAACCCTCACTAAAGGggataattttatgatcaccag-3′
> 5′- TAATACGACTCACTATAGGctactttggttggcggcaag-3′
> *cls-2* (R107.6):
> 5′-TAATACGACTCACTATAGGttcaaggaaaagttggacc-3′
> 5′- AATTAACCCTCACTAAAGGggtgcatttctgattccacc-3′
> *gei-17* (W10D5.3):
> 5′-AATTAACCCTCACTAAAGGTATGCTGATAATTTTGAACCGCT-3′
> 5′-TAATACGACTCACTATAGGTCATCAACAATAAGTCTATCATATGG-3′
> *hcp-1* (ZK1055.1):
> 5′-AATTAACCCTCACTAAAGGaagcgccagcaaaccgagtcgcc-3′
> 5′- TAATACGACTCACTATAGGgtcaatgtgacctttgacaggaagc-3′
> *hcp-2* (T06E4.1):
> 5′-TAATACGACTCACTATAGGtctcggaaaggaatcgaaaa-3′
> 5′-AATTAACCCTCACTAAAGGtcgttgtctccaattccaca-3′ *knl-1* (C02F5.1):
> 5′-TAATACGACTCACTATAGGttcacaaacttggaagccgctg-3′
> 5′- TAATACGACTCACTATAGGttcacaaacttggaagccgctg-3′
> *mdf-2* (Y69A2AR.30)
> 5′-TAATACGACTCACTATAGGTCAAAGGATCTGCCCAACTC-3′
> 5′-AATTAACCCTCACTAAAGGCGTCGAGAATGAGCGAAGTT-3′

## Embryonic viability assays

Embryonic viability assays were performed at 20 °C. Worms were singled onto plates 36 hr post-L4, upon recovery from dsRNA micro-injection. Worms were allowed to lay eggs for 12 hr before being removed from the laying plates. For each laying plate, the number of unhatched eggs, and of L1 larvae was counted. Plates were then left at 20 °C for another 36 hr before the total number of worms reaching L4/adulthood was counted. Control worms were analyzed with the same protocol with no preceding micro-injection. The proportion of viable progenies was calculated as the number of L4/adults divided by the number of eggs/L1.

## Live imaging and image analysis

All live and fixed acquisitions were performed on a Nikon Ti-E inverted microscope, equipped with a Yokogawa CSU-X1 (Yokogawa) spinning-disk confocal head with an emission filter wheel, using a Photometrics Scientific CoolSNAP HQ2 CCD camera. The power of 100 or 150 mW lasers was measured before each experiment with an Ophir VEGA Laser and energy meter. Fine stage control was ensured by a PZ-2000 XYZ Piezo-driven motor from Applied Scientific Instrumentation (ASI). The microscope was controlled with Metamorph 7 software (Molecular Devices).

For ex utero live imaging of embryos and one-cell zygotes, embryos were freed by dissecting worms on a cover slip in 6–8 µL meiosis medium *Laband et al., 2018*. Movies were acquired using a Nikon APO $\lambda$ S 60 x/1.40 oil objective and 2x2 binning. Images were acquired at 10, 15 or 20 second interval, over 2–4 Z-planes with a step size of 2 µm. Temperature was maintained at 23 °C using the CherryTemp controller system (Cherry Biotech).

Immunofluorescence was performed as described in *Gigant et al., 2017* using a 20 minutes cold (–20 °C) methanol fixation. Custom-produced Dylight 550-labeled (Thermo Scientific) rabbit anti-BUB-1

*Maton et al., 2015*, Dylight 650-labeled (Thermo Scientific) rabbit anti-CLS-2 *Maton et al., 2015*, and FITC-labeled mouse anti-α-tubulin (SIGMA F2168, DM1α) were used at a concentration of 1 µg/mL. DNA was stained with Hoechst at 2 µg/mL. Z-sections were acquired every 0.2 µm using a Nikon APO $\lambda$ S 100 x/1.45 oil objective. Maximum projections of relevant sections are presented.

Image and movie treatment, scaling and analysis were performed in FIJI (*Schindelin et al., 2012*). Protein signals at kinetochores were measured using linescans along the long axis of prometaphase I chromosomes. Background noise was measured using a linescan of similar length and width on the same Z-plane outside the spindle. Spindle area and fluorescence intensity of tubulin (microtubule density) were measured on sum projections of 4 Z-planes 45 s before anaphase I onset. For each sample, the background signal was measured using the same ROI placed at the center of the oocyte. Graphs represent ratios of the Integrated fluorescence/Background noise. To assay the effect of depleting BUB-1, HCP-1 and/or MDF-2 on the cell cycle timing and on meiotic spindle assembly and chromosome segregation, the time between the first frame of homologous chromosome visible separation (anaphase I) and the first frame of sister chromatid visible separation (anaphase II) was measured, and meiotic defects were assayed on the same movies. Spindles scored as 'disorganized' corresponded to visibly apolar or multipolar spindles.

## Western blotting

For each sample, gravid adult worms (24 hr post-L4) were washed in M9 buffer (22 mM $KH_2PO_4$, 42 mM $Na_2HPO_4$, 86 mM NaCl, and 1 mM $MgSO_4 \bullet 7H_2O$) supplemented with 0.1% Triton X100. Samples were then resuspended in 30 µL Laemmli buffer (1 X final) and incubated at 97 °C for 15 min, vortexed at 4 °C for 15 min, and boiled at 97 °C for 5 min. Worm extracts were loaded on NuPAGE 3–8% Tris Acetate gels (Invitrogen). Proteins were then transferred onto nitro-cellulose membranes which were incubated with 1 µg/µL primary antibodies in 1 X TBS-Tween (TBS-T) supplemented with 5% skim milk. Anti-BUB-1, anti-HCP-1 and anti-CLS-2 were custom-produced in rabbits, and affinity-purified against the antigen used for immunization (*Maton et al., 2015*). Mouse Anti-GFP (Roche 11814460001) was used to specifically detect the transgenes. Mouse anti-α-tubulin (Abcam Ab7291, DM1α) or rabbit custom-made anti-KLP-7 antibodies (*Gigant et al., 2017*), were used as loading controls. Blockings were performed in 1 X TBS-Tween +5% skim milk, and washes in 1 X TBS-T. Signals were revealed with HRP-coupled goat anti-mouse or anti-rabbit secondary antibodies (Jackson ImmunoResearch 115-035-003 or 11-035-003, respectively, 1:10,000 in 5% skim milk TBS-T). Revelation was performed on a Biorad ChemiDoc Imaging System.

## Yeast two hybrid assay

Full CDS or gene domains were cloned from wild type *C. elegans* (N2) cDNA. The yeast two hybrid assay was performed using the LexA-based system. In short, the HCP-1 KTD and CBD were fused to LexA binding domain in the bait pB27 plasmid, and other CDS sequences fused to Gal4 activating domain in the prey pP6 vector. Bait-encoding plasmids were transformed into L40Δga14 *S. cerevisiae* strain (MATα), prey plasmids into CG1945 strain (MAT a). Upon mating, diploid *S. cerevisae* were spotted on -Leu -Trp double-selection medium, and interactions were tested on -Leu -Trp -His medium. 5 mM of 3-amino-1,2,4-triazole (3AT) was added to abolish autoactivation by LexA::HCP-1(CBD). Image acquisitions were done on a Biorad ChemiDoc Imaging System.

## In silico protein sequence analyses

For comparison of TOGL domains among CLASPs (Ce: *Caenorhabditis elegans*, Dm: *Drosophila melanogaster*, Hs: *Homo sapiens*, Mm: *Mus musculus*, Xl: *Xenopus laevis*), protein sequence alignment and phylogenetic tree were generated using Clustal Omega (Version 1.2.4, https://www.ebi.ac.uk/Tools/msa/). The tree was rooted using the sequence of the TOG3 of human chTOG. The phylogenetic tree figure was generated using the FigTree software (v1.4.4, http://tree.bio.ed.ac.uk/software/figtree/).

Sequence alignment of the TOGL2 domains (At: *Arabidopsis thaliana*, Ce: *Caenorhabditis elegans*, Dm: *Drosophila melanogaster*, Hs: *Homo sapiens*, Mm: *Mus musculus*, Xl: *Xenopus laevis*) and the TOGL3 domain of *C. elegans*, or of the CLASPs CTDs (At: *Arabidopsis thaliana*, Ce: *Caenorhabditis elegans*, Ci: *Ciona intestinalis*, Dm: *Drosophila melanogaster*, Dr: *Danio rerio*, Gg: *Gallus gallus*, Hs: *Homo sapiens*, Mm: *Mus musculus*, Xl: *Xenopus laevis*) was done with MAFFT using the Snapgene software (https://www.snapgene.com/) and exported as Rich Text.

## Protein production and purification

Full-length BUB-1 and HCP-1 sequences were amplified from *C. elegans* cDNA and cloned in pFastBac dual expression vector in frame with a 6xHis tag. CLS-2 was cloned in pFastBac dual in frame with a C-terminal GFP tag, a TEV protease cleavage site, and a 6xHis tag. BUB-1, CLS-2::GFP and CLS-2$^{R970A}$::GFP production was performed in 2 L SF9 cells in Insect-XPRESS (Lonza) medium (1x10^6 cells/mL), infected with amplified baculovirus for 48 hr at 27 °C. HCP-1 was produced similarly but in Hi-5 cells maintained in EX-Cell 405 medium (Sigma-Aldrich) and collected after 66 hr infection. Cells were harvested by centrifugation at 700xg and resuspended in 10–15 mL lysis buffer (150 mM KCl, 1 mM MgCl$_2$, 0.1% Tween-20, protease inhibitor (Complete EDTA-free tablets, Roche)) per liter of culture, supplemented with 5% glycerol for CLS-2::GFP and CLS-2$^{R970A}$::GFP. BUB-1 was purified in 25 mM HEPES pH 7.2, HCP-1 in 50 mM MES pH 6.5, and CLS-2::GFP and CLS-2$^{R970A}$::GFP in 50 mM PIPES pH 6.8 supplemented with 50 mM glutamate and 50 mM arginine to increase protein solubility and purification yield (*Lawrence et al., 2018*; *Golovanov et al., 2004*). Cells were homogenized in a Dounce homogenizer and lysed by sonication for 30 s at 50% amplitude using a 6 mm diameter probe. Lysates were clarified by ultracentrifugation at 100,000xg for 1 hr at 4 °C. Sample loading, column washes and elution were performed using an ÄKTA Pure chromatography system (Cytiva). Supernatants were loaded on 1 mL HiTrap TALON crude (Cytiva) columns equilibrated with Buffer A (150 mM KCl, 1 mM MgCl$_2$). The columns were washed with 10 column volumes (CVs) of buffer A followed by 30 CVs of buffer A' (150 mM KCl, 1 M MgCl$_2$). Columns were then equilibrated with 20 CVs of buffer A. Elution was performed with 30 CVs of a gradient from 0 to 100% buffer B (150 mM KCl, 1 mM MgCl$_2$, 300 mM Imidazole). A, A' and B buffer pHs were adjusted to protein-specific lysis buffer pH. Absorbance was measured at 280 nm. Fractions were analyzed by SDS-PAGE and Coomassie staining. Fractions containing pure proteins were pooled, concentrated using Amicon Ultra-15 centrifugation units and desalted against buffer A using Econo-Pac 10DG Desalting columns (Bio-Rad). Proteins were aliquoted, snap frozen in liquid nitrogen and stored at –80 °C. CLS-2::GFP and CLS-2$^{R970A}$::GFP protein were further purified using a gel filtration column Superdex 200 Increase 10/300 GL (Cytiva). Equilibration, loading and isocratic elution were made at a flow rate of 0.4 mL/min with 25 mM HEPES pH 7.2, 150 mM KCl, 1 mM EGTA, and 1 mM MgCl$_2$.

CLS-2 TOGL2 (aa1-276) and TOGL3 (aa441-699) domains were expressed in *E. coli* Rosetta2 cells using pProEx-HTb 6xHis expression vectors. Transformed bacteria were allowed to reach 0.4 OD600nm at 37 °C. The cultures were then transferred to 20 °C and expression was induced overnight (~18 hr) with 0.1 mM IPTG. Bacteria were harvested by centrifugation at 4000xg for 20 min resuspended and washed with 1 x PBS by centrifugation. Cells were resuspended in lysis buffer (25 mM MOPS pH 7.2, 300 mM NaCl, 10 mM ß-mercaptoethanol) supplemented with protease inhibitors (Complete EDTA-free tablets, Roche). Cells were lysed by sonication using a 6 mm diameter probe at 50% amplitude for 2 min. The lysates were clarified by ultracentrifugation at 90,000xg for 1 hr at 4 °C and loaded on 1 mL HisTrap Excel columns (Cytiva). Purification of CLS-2 domains was performed using the same procedure as BUB-1, HCP-1 and CLS-2::GFP (see previous paragraph) but using different composition for buffer A (25 mM MOPS pH 7.2, 300 mM NaCl, and 10 mM ß-mercaptoethanol), A' (25 mM MOPS pH 7.2, 1 M NaCl, and 10 mM ß-mercaptoethanol) and B (25 mM MOPS pH7.2, 300 mM NaCl, and 10 mM ß-mercaptoethanol, 300 mM imidazole). CLS-2 domains were dialyzed against 25 mM MOPS pH 7.4, 1 mM EGTA, 300 mM KCl, and 10 mM ß-mercaptoethanol, frozen in liquid nitrogen and stored at –80 °C.

Tubulin was purified from pig brains following high salt protocol and cycles of polymerization and depolymerization as in *Castoldi and Popov, 2003*. Tubulin was then labeled with either NHS-ester-ATTO 565 (ATTO-TEC), NHS-ester-ATTO 488 (ATTO-TEC) or EZ-Link Sulfo NHS-LC-LC-Biotin (ThermoFisher #21338). Labeling dyes or linkers were removed by two cycles of polymerization/depolymerization *Hyman et al., 1991*. In brief, unlabeled polymerized tubulin was incubated 40 min at 37 °C in the presence of 5 mM of succinimidyl ester-coupled reagent in labeling buffer (0.1 M HEPES pH 8.6, 1 mM MgCl$_2$, 1 mM EGTA, and 40% glycerol (volume/volume)). Microtubules were then spun down through a low pH cushion (60% glycerol, 1 x BRB (80 mM K-PIPES pH 6.8, 1 mM MgCl$_2$, and 1 mM EGTA)), resuspended in 50 mM K-glutamate pH 7.0, 0.5 mM MgCl$_2$, and left to depolymerized on ice for 30 min in a small glass Dounce homogenizer. Depolymerized labeled tubulin was recovered from a 120,000xg centrifugation at 2 °C and resuspended in 4 mM MgCl$_2$, 1 mM GTP, 1 x BRB. An additional cycle of polymerization was performed at 37 °C for 40 min. Microtubules were sedimented

at 120,000xg for 20 min at 37 °C and depolymerized in ice-cold 1 x BRB buffer. Soluble tubulin was recovered from a 10 min 150,000xg centrifugation at 2 °C, diluted to 15–20 mg/mL in 1 x BRB, aliquoted, frozen in liquid nitrogen and stored at –80 °C.

### Protein-protein interactions by size exclusion chromatography

To assess the interaction of the CLS-2 domains with soluble tubulin, an equimolar mix was made at a concentration of 25 µM of each protein in equilibration buffer (25 mM HEPES pH 7.0, 80 mM KCl, 1 mM EGTA, 1 mM $MgCl_2$, and 5% glycerol). Samples (50 µL) were loaded on a Superdex 200 Increase 10/300 GL (Cytiva) at a flow rate of 0.4 mL/min. Proteins were followed by measuring the absorbance at 280 nm and peaking fractions were loaded and analyzed by SDS-PAGE and Coomassie staining.

### Microtubules bundling assays

A 100 µL mixture of 80 µM unlabeled and ATTO-565-labeled tubulin (12:1 ratio) was incubated 5 min at 4 °C and centrifuged at 100,000xg for 10 min to remove aggregated labeled tubulin. The supernatant was left to polymerize at 35 °C for 30 min in 1 mM GTP, 1 x BRB buffer for 30 min. An equal volume of 1 x BRB buffer with 20 µM docetaxel (Sigma-Aldrich) was added, and the reaction was further incubated for 15 min at room temperature before being centrifuged at 50,000xg for 10 min at 25 °C. The supernatant was discarded, the pellet was gently washed with a volume of warm (35 °C) 10 µM docetaxel, 1 x BRB. The pellet was rehydrated by incubating for 10 min at room temperature in a volume of 10 µM docetaxel, and then resuspended by pipetting up and down. The stabilized microtubule solution was diluted in 10 µM docetaxel,1x BRB to a final tubulin concentration of ~160 nM to facilitate visualization under the fluorescent microscope. The diluted microtubule suspension was incubated 5 min at room temperature with the protein of interest at a concentration of 200 nM to 250 nM in a microtube. The mixture was transferred in a ~10 µL microchamber between a microscope slide and a coverslip assembled with thin strips of double-sided tape. The chamber was immediately imaged with a spinning-disk confocal. An image of a single focal plane was captured at ×60 magnification with a 1.4 N.A. oil immersion objective, using 561 nm excitation laser.

### Microtubules pelleting assays

Stabilized microtubules were prepared as stated above except that no labeled tubulin was added. Final reaction volumes of 50–100 µL of 1 µM purified BUB-1 and 0–1.7 µM stabilized microtubules were prepared in 10 µM docetaxel, 80 mM KCl, 1 x BRB buffer, into ultracentrifugation microtubes. Mixtures were incubated 15 min at room temperature and centrifuged at 50,000xg for 10 min. The supernatants were kept in 1 x Laemli sample buffer (LSB) from a 5 x solution (400 mM TRIS-HCl pH6.8, 450 mM DTT, 10% SDS, 50% glycerol, and 0.006% w/vol bromophenol blue). Pellets were washed with 10 µM docetaxel in 1 x BRB and directly resuspended in 1 x LSB. Supernatants and pellets were analyzed by SDS-PAGE (10% acrylamide) and Coomassie staining.

### In vitro microtubule dynamics

Biotinylated GMPCPP-stabilized microtubule seeds were obtained by mixing biotinylated-tubulin and fluorescent tubulin (ATTO488-tubulin) at a 4:1 ration to a 10 µM final concentration of tubulin. This mix was incubated at 37 °C in 1 x BRB supplemented with 0.5 µM GMPCPP (Jenabioscience). After a 1 hr incubation, docetaxel (Sigma-Aldrich) was added to a final concentration of 1 µM and the reaction was incubated for 30 min at 30 °C. Microtubule seeds were pelleted at 100,000 g for 10 min at 25 °C. The pellet was resuspended in 1 x BRB, 0.5 mM GMPCPP, and 1 µM docetaxel. Seeds were aliquoted, frozen in liquid nitrogen and stored at –80 °C in cryotubes for up to 3 weeks.

Single microtubule dynamics assays were performed in a ~20 µL flow chamber between a glass slide and a coverslip assembled with double-sided sticky tape. Glass slides were cleaned and passivated using a protocol adapted from *Aumeier et al., 2016*. In brief, slides were washed successively in water, acetone, ethanol, 2% Hellmanex detergent for at least 30 min in each reagent in a glass beaker immersed into an ultrasonic bath. Glass slides were then treated with a 1 mg/mL PEG-silane solution (MW 30 K; Creative PEG works) in 96% ethanol, 0.1% HCl. In addition, coverslips were plasma-cleaned for 2 min before being treated with PEG-silane-biotin (MW 10 K Creative PEG works) overnight at room temperature under gentle agitation. Slides and coverslips were abundantly washed with large volumes of MilliQ-water and dried using pressurized air blowing. Dried slides and coverslips were

stored at 4 °C for a maximum of 3 weeks in clean plastic boxes. Prior to use, a microchamber of around 10 µL in volume was fabricated using a slide, a coverslip and double-sided tape (~70 µm thickness). A total of 100 µL of 50 µg/mL neutravidin (Invitrogen A2666), BRB-BSA (1 x BRB, 0.2% BSA) solution were flowed into the chamber. The chamber was then kept at room temperature (22–23°C) and never allowed to dry. The neutravidin solution was incubated for 5 min in the chamber. The chamber was washed twice for 1 min with 100 µL of 0.1 mg/mL PLL-PEG (PLL20k-G35-PEG2k, Jenkem), and a third time with 300–400 µL of BRB-BSA. A solution of GMPCPP-stabilized microtubule seeds diluted in BRB-BSA was flowed into the chamber and incubated for 3 min. Excess of seeds was then removed by several washes of >300 µL BRB-BSA before introduction of the elongation mix containing 12 µM total tubulin (94% unlabeled pig brain tubulin, 6% labeled pig brain tubulin), in 40 mM PIPES pH6.8, 10 mM HEPES pH 7.5, 44 mM KCl, 5 mM MgCl$_2$, 1.5 mM EGTA, 0.2% methylcellulose (1500 cP), 4 mM DTT, 1 mM GTP, 0.5 mM ATP buffer supplemented with 128 nM catalase, 500 nM glucose oxidase and 40 mM glucose antifading agents, and in the presence of 100 nM freshly thawed BUB-1, HCP-1 and/ or CLS-2::GFP. Microtubule dynamics was monitored between 22.5°C and 23°C on an azimuthal TIRF microscope (Nikon Eclipse Ti2 equipped with the Ilas2 module, Gataca systems) using a 60 x, 1.47NA oil immersion TIRF objective and a Photometrics Prime BSI sCMOS camera. Two-channel acquisitions (488 nm and 561 nm) were performed every 3 s for 20 min. Imaged were acquired at 50 ms exposure.

## Quantification of microtubule dynamics

Microtubule dynamics parameters were extracted from kymographs generated from TIRF microscopy image sequences using the built-in plugin 'Multi Kymographs' in FIJI or 'MultipleKymograph' in ImageJ. A macro was used to generate kymograph from each microtubule using multiple 'Segmented Line' ROIs (region of interests). Parameters were then extracted manually by measuring rates (slopes) and durations (vertical distances) on kymographs (*Zwetsloot et al., 2018*). During our experiments (especially in the presence of the full BHC module), microtubules often displayed complex behaviors, such as a different growth rates during a single growth excursion with intercalated pause events and without undergoing catastrophe (e.g. growth to pause to regrowth). Because microtubule dynamics can be influenced by microtubule age and lattice effects (*Odde et al., 1995*; *Gardner et al., 2011*; *Rai et al., 2021*), it is a better approach to consider the lifetime history of single microtubules to quantify catastrophe and rescue events (*Zanic, 2016*). Lifetime refers here to the period of time from which a microtubule emanates from the seed, to the time it starts shrinking all the way to the seed. The rescue frequency was measured as the number of rescues observed per microtubule, divided by the total time it spent shrinking within its lifetime. Similarly, catastrophe frequency corresponds here to the number of observed catastrophes per microtubule, divided by the total duration of growth excursion(s) of this microtubule, even if the elongation periods are interspersed with pause events, but excluding phases of shrinkage. The percentage of time spent in pause corresponds the duration of all pause events for a single microtubule divided by its lifetime. Measurements are available in *Figure 6—source data 1* – Panel B-F source data.

## Figure preparation, graphs and statistical analyses

Figures and illustrations were done in the Affinity Designer software (ver. 1.10.3). Graphical representation of data and statistical analyses were performed using the GraphPad Prism software (ver. 8.4.3 (471)). Statistical tests used are specified in the corresponding figure legends and their source data file.

## Acknowledgements

We thank all members of the Pintard and Dumont labs for support and advice. We are grateful to Patricia Moussounda, Téo Bitaille and Vincent Maupu-Massamba for providing technical support. We thank Dhanya Cheerambathur for worm strains and Benoit Palancade for yeast two hybrid strains. We thank Jérémie Gaillard and members of the CytoMorphoLab for technical advices and assistance with TIRF microscopy and microtubule-based assays. Some strains were provided by the CGC, which is funded by NIH Office of Research Infrastructure Programs (P40 OD010440). N Macaisne was supported by an ANR contract (grant ANR-19-CE13-0015), L Bellutti was supported by a post-doctoral fellowship from the Fondation pour la Recherche Médicale (FRM). This work was supported by CNRS and University Paris Cité, by NIH R01GM117407 and R01GM130764 (JC.

Canman), and by grants from the European Research Council consolidator grant (ERC-CoG) ChromoSOMe grant 819179 and from the Agence Nationale de la Recherche ANR-19-CE13-0015 (J Dumont).

## Additional information

### Funding

| Funder | Grant reference number | Author |
|---|---|---|
| Agence Nationale de la Recherche | ANR-19-CE13-0015 | Nicolas Macaisne |
| Fondation pour la Recherche Médicale | Post Doctoral Fellowship | Laura Bellutti |
| European Research Council | CoG Chromosome 819179 | Julien Dumont |
| Agence Nationale de la Recherche | ANR-19-CE13-0015 | Julien Dumont |
| National Institutes of Health | R01GM117407 | Julie C Canman |
| National Institutes of Health | R01GM130764 | Julie C Canman |

The funders had no role in study design, data collection and interpretation, or the decision to submit the work for publication.

### Author contributions

Nicolas Macaisne, Conceptualization, Resources, Data curation, Formal analysis, Investigation, Visualization, Methodology, Writing - original draft, Writing - review and editing; Laura Bellutti, Kimberley Laband, Conceptualization, Formal analysis, Investigation, Visualization, Methodology; Frances Edwards, Conceptualization, Resources, Formal analysis, Investigation, Visualization; Laras Pitayu-Nugroho, Formal analysis, Investigation, Visualization; Alison Gervais, Resources, Formal analysis, Investigation, Visualization; Thadshagine Ganeswaran, Hélène Geoffroy, Resources; Gilliane Maton, Conceptualization, Resources; Julie C Canman, Conceptualization, Resources, Writing - original draft, Writing - review and editing; Benjamin Lacroix, Conceptualization, Resources, Formal analysis, Investigation, Writing - original draft, Writing - review and editing; Julien Dumont, Conceptualization, Resources, Data curation, Supervision, Funding acquisition, Methodology, Writing - original draft, Project administration, Writing - review and editing

### Author ORCIDs

Nicolas Macaisne http://orcid.org/0000-0002-0109-9845
Kimberley Laband http://orcid.org/0000-0002-8535-2050
Thadshagine Ganeswaran http://orcid.org/0000-0003-4986-4419
Hélène Geoffroy http://orcid.org/0000-0002-3231-8369
Julien Dumont http://orcid.org/0000-0001-5312-9770

### Decision letter and Author response

Decision letter https://doi.org/10.7554/eLife.82579.sa1
Author response https://doi.org/10.7554/eLife.82579.sa2

## Additional files

### Supplementary files
• MDAR checklist
• Supplementary file 1. List of *C. elegans* strains used in this study.

## Data availability

All data generated or analysed during this study are included in the manuscript and supporting files; source data files have been provided for all figures.

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
