## [Editor Report]

This paper on the regulation of microtubule dynamics during *C. elegans* meiosis presents important findings that will be of interest to scientists in the broad field of microtubule function in both mitosis and meiosis. The experiments are beautifully conducted and presented and support the conclusions of the paper in a compelling manner. The results are interesting and add to our understanding of the control of microtubule dynamics at the kinetochore and its functional consequences for meiosis.

---

## [Decision Letter]

**Decision letter after peer review:**

Thank you for submitting your article "Synergistic stabilization of microtubules by BUB-1, HCP-1 and CLS- 2 controls meiotic spindle assembly in *C. elegans* oocytes" for consideration by *eLife*. Your article has been reviewed by 3 peer reviewers, and the evaluation has been overseen by a Reviewing Editor and Anna Akhmanova as the Senior Editor. The reviewers have opted to remain anonymous.

Essential revisions:

1) Some of the observations made in cells after BHC module modifications indicate that the spindle assembly checkpoint may be activated, which could necessitate modifying some of the conclusions. Testing whether the checkpoint is indeed activated in these experiments or not would be important.

2) To clarify whether the effects observed in vitro with purified proteins indeed reflect the activity of the intact BHC module in cells, the analysis of binding-defective mutants in vitro would be very informative. Specifically, including HCP-1 mutants defective in CLS-2 and/or BUB-1 binding would help determine whether the enhancement of microtubule pausing that is observed in the presence of all three components requires assembly of the module.

3) The discussion should be improved with the goal to better explain differences and similarities with mitosis and meiosis in systems other than *C. elegans*. The authors could also use this opportunity to better highlight to which extent their work here presents a conceptual advance over previous knowledge in the field.

*Reviewer #1 (Recommendations for the authors):*

Figure 3. A redundant role in chromosome segregation looks clear, but other aspects are less convincing. Either the legend or the method section should contain information on how "disorganized spindle" is defined. A statistical analysis is required to compare between Knl-1 (RNAi) and knl-1 gel-17 (RNAi), in the presence of KNLd85-505, for both spindle density and the spindle size. There appears to be no difference in the spindle density between these two in the quantification (B), but the images shown in (A) give a different impression. Representative images should be selected.

L304. "CLS-2 functions through a single TOGL domain" is a misleading statement. It sounds like TOGL2 alone is sufficient function. Maybe "The CLS-2 function does not require TOGL3".

*Reviewer #2 (Recommendations for the authors):*

– Lines 208 – 210: "In most species, CLASPs targeting to their various subcellular localizations, including to kinetochores, relies on interactions with various adapter proteins via a C-terminal domain (CTD)".

More than one reference is required here to support this statement.

– Lines 213 – 214: "Both transgenes are expressed at similar levels".

Refer to the Western blot in Figure 5 —figure supplement 1D.

– Lines 377 – 380: "reconstituting the BHC module in vitro displays both additive (promotion of the growth rate and rescue frequency, and inhibition of catastrophe) and synergistic (promotion of pause) effects on microtubule dynamics compared to individual components, which leads to microtubule stabilization."

There are two issues with this statement: (1) there is no evidence that adding all three components actually results in complex formation (the same claim is made in Line 462); and (2) the effect on rescue frequency and inhibition of catastrophe is not "additive", since the effect is no different from that observed with CLS-2 alone.

– Lines 467 – 470: "Meiotic spindle assembly requires an intact BHC module and the strong microtubule pause promoting effect of the module, while chromosome segregation and polar body extrusion only depend on the rescue promoting and catastrophe reducing effects of CLS-2 alone."

As mentioned in the public review, the idea that microtubule pausing is promoted by the intact module is appealing but the requirement for module assembly needs to be demonstrated in vitro by using binding defective mutants.

– Line 488 – 492: "Therefore, unlike mammalian, or yeasts and *Drosophila* CLASPs, which respectively prevent catastrophe and promote rescue by concentrating at a region located behind the outmost microtubule end, or by accumulating at speckles along the lattice, the stabilizing effect of *C. elegans* CLS-2 does not require its plus-end nor lattice speckled accumulation. Instead binding to HCP-1 and BUB-1 could modulate CLS-2 effect on microtubules."

This statement is in apparent contradiction to the results: addition of CLS-2 alone already promotes rescue and lowers the catastrophe frequency in vitro, so CLS-2 does not need to be part of the BHC module.

– Graphs: describe what the error bars represent in the figure legends.

– Figure 2D, E: for clarify, it would be good to already indicate in these panels which HCP-1 constructs correspond to the NTD and CTD deletions, which are then analyzed in the following panels.

– Figure 5: The characterization of the CLS-2 CTD deletion mutant would fit better in Figure 4B (the mutant is already shown in the cartoon in Figure 4A). This would more clearly separate analysis of BHC integrity (Figure 4) from analysis of CLS-2's microtubule binding region (Figure 5).

– Figure 6 —figure supplement 1D should already be mentioned in the Results section, not only in the discussion.

– Movie 5: The cls-2(RNAi) condition is worse than the cls-2(RNAi) + hcp-1(RNAi) condition: are the conditions mis-labelled?

*Reviewer #3 (Recommendations for the authors):*

The discussion of the data would benefit from some editing. The data obtained in this study are compared to published data on the mammalian homologues, and differences are discussed without fully considering that the mammalian data were obtained from experiments on somatic cells, not cells undergoing meiosis. Some of the differences may therefore be due to differences between meiosis and mitosis, not the species.

---

## [Author Response]

Essential revisions:1) Some of the observations made in cells after BHC module modifications indicate that the spindle assembly checkpoint may be activated, which could necessitate modifying some of the conclusions. Testing whether the checkpoint is indeed activated in these experiments or not would be important.

We have included a new set of experiments to test this assumption and we show that SAC activation is not causative of the phenotypes observed following BHC module perturbation.

2) To clarify whether the effects observed in vitro with purified proteins indeed reflect the activity of the intact BHC module in cells, the analysis of binding-defective mutants in vitro would be very informative. Specifically, including HCP-1 mutants defective in CLS-2 and/or BUB-1 binding would help determine whether the enhancement of microtubule pausing that is observed in the presence of all three components requires assembly of the module.

We have identified a point mutation in CLS-2 (R970A), which precludes binding to HCP-1 and therefore disrupts BHC module integrity. We have included in vivo and in vitro analysis of this mutant to show that microtubule pausing enhancement requires an intact BHC module.

3) The discussion should be improved with the goal to better explain differences and similarities with mitosis and meiosis in systems other than *C. elegans*. The authors could also use this opportunity to better highlight to which extent their work here presents a conceptual advance over previous knowledge in the field.

We have edited the discussion accordingly. We have additionally conducted some experiments that were not required by the Reviewers. These new results provide a quantitative analysis of BUB-1 and BHC module component levels at kinetochores in the absence of the BUB-1 binding protein BUB-3, and further define the minimal HCP-1 domain required for kinetochore localization. New experiments include 1/ quantitative western-blot analysis of BUB-1 protein level, and HCP-1 and CLS-2 localizations, in the *bub-3*D *mutant*, 2/ further refinement of the HCP-1 sequence required for its kinetochore localization (from 232aa to a short stretch of 75aa corresponding to the HCP-1^D1311-1386^ deletion mutant) by generation and in vivo analysis of 4 new HCP-1 deletion mutants and a C-terminally GFP-tagged full-length HCP-1.

Reviewer #1 (Recommendations for the authors):Figure 3. A redundant role in chromosome segregation looks clear, but other aspects are less convincing. Either the legend or the method section should contain information on how "disorganized spindle" is defined.

We have added a short description of the “disorganized spindle” scoring in the method section. Briefly, disorganized spindles corresponded to visibly apolar or multipolar spindles.

A statistical analysis is required to compare between Knl-1 (RNAi) and knl-1 gel-17 (RNAi), in the presence of KNLd85-505, for both spindle density and the spindle size.

This statistical analysis has been added in the revised version of our manuscript.

There appears to be no difference in the spindle density between these two in the quantification (B), but the images shown in (A) give a different impression. Representative images should be selected.

This has been corrected in the revised version of our manuscript.

L304. "CLS-2 functions through a single TOGL domain" is a misleading statement. It sounds like TOGL2 alone is sufficient function. Maybe "The CLS-2 function does not require TOGL3".

This statement originated from similar findings made by the Akhmanova lab on human CLASP2 (Aher et al., 2018). We have nevertheless edited the revised version of our manuscript according to Reviewer 1 suggestion.

Reviewer #2 (Recommendations for the authors):– Lines 208 – 210: "In most species, CLASPs targeting to their various subcellular localizations, including to kinetochores, relies on interactions with various adapter proteins via a C-terminal domain (CTD)".More than one reference is required here to support this statement.

Additional relevant references have been added to the revised version of our manuscript.

– Lines 213 – 214: "Both transgenes are expressed at similar levels".Refer to the Western blot in Figure 5 —figure supplement 1D.

Done.

– Lines 377 – 380: "reconstituting the BHC module in vitro displays both additive (promotion of the growth rate and rescue frequency, and inhibition of catastrophe) and synergistic (promotion of pause) effects on microtubule dynamics compared to individual components, which leads to microtubule stabilization."There are two issues with this statement: (1) there is no evidence that adding all three components actually results in complex formation (the same claim is made in Line 462);

We did not intend to mean that BHC components form a complex. This is why we termed it “module” and not “complex”. In fact, we now added new in vitro experiments analyzing a CLS2 mutant that cannot bind HCP-1 (CLS-2^R970A^, see below). We found that the synergistic effect of BHC components on microtubule pausing is lost when this mutant is used instead of wildtype CLS-2. This suggests that binding of HCP-1 and CLS-2, and the presence of BUB-1, are essential for microtubule pausing to be synergistically increased. Although we strongly suspect that BUB-1 directly binds to HCP-1, we were unable to demonstrate direct physical interaction between the BUB-1 Kinase Domain and HCP-1 KTD (Kinetochore Targeting Domain) using a Y2H assay, and we therefore cannot definitively conclude on that specific point.

and (2) the effect on rescue frequency and inhibition of catastrophe is not "additive", since the effect is no different from that observed with CLS-2 alone.

We apologize for not being clearer about this point. We in fact meant that the additive effects of HCP-1 (increase of the microtubule growth rate, not observed with CLS-2 alone) and CLS-2 (increase of rescue and decrease of catastrophe) are observed when both components (or a combination of all 3 BHC components) are combined. We have modified the text to make this point clearer in the revised version of our manuscript.

– Lines 467 – 470: "Meiotic spindle assembly requires an intact BHC module and the strong microtubule pause promoting effect of the module, while chromosome segregation and polar body extrusion only depend on the rescue promoting and catastrophe reducing effects of CLS-2 alone."As mentioned in the public review, the idea that microtubule pausing is promoted by the intact module is appealing but the requirement for module assembly needs to be demonstrated in vitro by using binding defective mutants.

We thank Reviewer 2 for raising this important point. We have added a new set of in vivo and in vitro experiments that specifically address this concern and these new results have been incorporated in the revised version of our manuscript. By comparing CLASP CTD (C-terminal Domain) sequences across species, we identified a highly conserved stretch of 3 amino-acids (VRK) at the C-terminus of all CTDs. By mutating the central Arginine residue into an Alanine (R970A mutant) in the CTD of CLS-2, we showed that it is essential for direct binding to the HCP-1 CBD (CLS-2 Binding Domain; by Y2H analysis), for kinetochore localization of CLS-2 (by analyzing in vivo oocytes of the corresponding mutant), and that it phenocopied the full CLS2 CTD deletion mutant (by analyzing spindle formation and chromosome segregation in live oocytes). The CLS-2^R970A^ mutant is thus unable to bind to HCP-1 and hence precludes BHC module integrity. We then purified the corresponding recombinant protein in insect cells and performed a new set of in vitro TIRF-based assays using this mutant alone, or combined with HCP-1 and BUB-1 proteins. Strikingly, although the CLS-2^R970A^ mutant alone had identical effects on microtubule dynamics as wild-type CLS-2 (decrease of microtubule growth rate and catastrophe frequency, and increase of the rescue frequency), it was unable to induce the strong microtubule pausing enhancement induced by wild-type CLS-2 when combined with HCP-1 and BUB-1. Altogether, we conclude that an intact BHC module is essential for the synergistic enhancement of microtubule pausing observed when BHC module components are combined.

We have also revised the method for analysis and statistical comparisons of all our in vitro microtubule-based assays. While we previously analyzed and compared experiments, which limited the statistical power of our comparisons because of the limited number of individual experiments (n=3 for each condition), we now analyze and compare individual events. This is in line with previous work on microtubule dynamics in vitro (Rai et al., 2021; Zanic, 2016). This was also motivated by the fact that, in our experiments (especially in the presence of the full BHC module), microtubules often display complex behaviors, such as a different growth rates during a single growth excursion with intercalated pause events and without undergoing catastrophe (e.g., growth to pause to regrowth). Because microtubule dynamics can be influenced by microtubule age and lattice effects (Gardner et al., 2011; Odde et al., 1995; Rai et al., 2021), we think it is a better approach to consider the lifetime history of single microtubules to quantify catastrophe and rescue events (Zanic, 2016). Lifetime refers here to the period of time from which a microtubule emanates from the seed, to the time it starts shrinking all the way to the seed. The rescue frequency is measured as the number of rescues observed per microtubule, divided by the total time it spent shrinking within its lifetime. Similarly, catastrophe frequency corresponds here to the number of observed catastrophes per microtubule, divided by the total duration of growth excursion(s) of this microtubule, even if the elongation periods are interspersed with pause events, but excluding phases of shrinkage. We average frequencies of individual microtubules (instead of dividing the total number of events by the total growth duration of all microtubules as in the original version of our manuscript). This calculation method returns a harmonic mean, which minimizes the bias effect of long polymerization periods (low catastrophe), especially when microtubules do not experience catastrophe during the time of observation (0 catastrophe). This explains why absolute values of catastrophe and rescue frequencies are overall higher in the revised Figure 6D and E. Importantly, this does not affect the relative differences between conditions. Finally, we included new experiments and thus we increased the number of values. This explains why the difference in rescue frequencies between CLS-2 alone or in combination with HCP-1 is now statistically significant. The new ‘single-event’ calculation method of microtubule dynamics also gives more weight to the repeated short depolymerization events (high recue frequency) observed when CLS-2 is combined to HCP-1 (kymographs in Figure 6B). Overall, this new quantification approaches dramatically increased the statistical power of all the analyses we performed.

– Line 488 – 492: "Therefore, unlike mammalian, or yeasts and Drosophila CLASPs, which respectively prevent catastrophe and promote rescue by concentrating at a region located behind the outmost microtubule end, or by accumulating at speckles along the lattice, the stabilizing effect of *C. elegans* CLS-2 does not require its plus-end nor lattice speckled accumulation. Instead binding to HCP-1 and BUB-1 could modulate CLS-2 effect on microtubules."This statement is in apparent contradiction to the results: addition of CLS-2 alone already promotes rescue and lowers the catastrophe frequency in vitro, so CLS-2 does not need to be part of the BHC module.

We apologize for what is clearly a grammatical mistake. We should not have linked these two sentences. We have edited the text accordingly to avoid giving impression that we conclude on the effect of CLS-2, in preventing catastrophe while promoting rescue, be dependent on HCP-1 or BUB-1.

– Graphs: describe what the error bars represent in the figure legends.

Done.

– Figure 2D, E: for clarify, it would be good to already indicate in these panels which HCP-1 constructs correspond to the NTD and CTD deletions, which are then analyzed in the following panels.

Done. Also, to avoid confusion between the HCP-1 and CLS-2 CTDs, we have now renamed the HCP-1 CTD into HCP-1 KTD (Kinetochore Targeting Domain), and the HCP-1 NTD into HCP-1 CBD (CLS-2 Binding Domain).

– Figure 5: The characterization of the CLS-2 CTD deletion mutant would fit better in Figure 4B (the mutant is already shown in the cartoon in Figure 4A). This would more clearly separate analysis of BHC integrity (Figure 4) from analysis of CLS-2's microtubule binding region (Figure 5).

We respectfully disagree with this suggestion as we think it is better to present all the CLS-2 structure/function mutants together in a single figure. However, we agree with Reviewer 2 that displaying this mutant in the cartoon of Figure 4b was misleading and for clarity, we have removed the CLS-2 CTD mutant from this cartoon.

– Figure 6 —figure supplement 1D should already be mentioned in the Results section, not only in the discussion.

We again respectfully disagree. In fact, the “LxxPTPh”-like motif we identified in CLS-2 diverges drastically (LPKRPTPQ) from the canonical motif. It is in fact most likely not a bona fide “LxxPTPh” motif. We only included analysis of this mutant to exclude any doubt about CLS-2 potentially interacting with EB proteins and thus concentrating at microtubule (+)-ends. We do not think this is a major finding of our present study and therefore that it does not deserve to be included in the main figures.

– Movie 5: The cls-2(RNAi) condition is worse than the cls-2(RNAi) + hcp-1(RNAi) condition: are the conditions mis-labelled?

We apologize for this bad choice of samples. We have edited these movies to include more representative examples of the corresponding phenotypes.

Reviewer #3 (Recommendations for the authors):The discussion of the data would benefit from some editing. The data obtained in this study are compared to published data on the mammalian homologues, and differences are discussed without fully considering that the mammalian data were obtained from experiments on somatic cells, not cells undergoing meiosis. Some of the differences may therefore be due to differences between meiosis and mitosis, not the species.

The discussion has been edited according to Reviewer 3’s suggestion. We note that although our manuscript primarily focuses on oocyte meiotic divisions, we systematically included analyses of the various mutants in *C. elegans* zygotes during mitosis and obtained identical results. We therefore don’t think that the differences we highlight in the present study are due to differences between meiosis and mitosis, but rather to species-specific variations, which we discuss extensively throughout the Discussion section of our manuscript.

References:

Aher, A., M. Kok, A. Sharma, A. Rai, N. Olieric, R. Rodriguez-Garcia, E.A. Katrukha, T. Weinert, V. Olieric, L.C. Kapitein, M.O. Steinmetz, M. Dogterom, and A. Akhmanova. 2018. CLASP Suppresses Microtubule Catastrophes through a Single TOG Domain. *Dev Cell*. 46:4058 e48.

Essex, A., A. Dammermann, L. Lewellyn, K. Oegema, and A. Desai. 2009. Systematic analysis in *Caenorhabditis elegans* reveals that the spindle checkpoint is composed of two largely independent branches. *Molecular biology of the cell*. 20:1252-1267.

Gardner, M.K., M. Zanic, C. Gell, V. Bormuth, and J. Howard. 2011. Depolymerizing kinesins Kip3 and MCAK shape cellular microtubule architecture by differential control of catastrophe. *Cell*. 147:1092-1103.

Kim, T., P. Lara-Gonzalez, B. Prevo, F. Meitinger, D.K. Cheerambathur, K. Oegema, and A. Desai. 2017. Kinetochores accelerate or delay APC/C activation by directing Cdc20 to opposing fates. *Genes Dev*. 31:1089-1094.

Kim, T., M.W. Moyle, P. Lara-Gonzalez, C. De Groot, K. Oegema, and A. Desai. 2015. Kinetochore-localized BUB-1/BUB-3 complex promotes anaphase onset in *C. elegans*. *J Cell Biol*. 209:507-517.

Odde, D.J., L. Cassimeris, and H.M. Buettner. 1995. Kinetics of microtubule catastrophe assessed by probabilistic analysis. *Biophys J*. 69:796-802.

Rai, A., T. Liu, E.A. Katrukha, J. Estevez-Gallego, S.W. Manka, I. Paterson, J.F. Diaz, L.C. Kapitein, C.A. Moores, and A. Akhmanova. 2021. Lattice defects induced by microtubulestabilizing agents exert a long-range effect on microtubule growth by promoting catastrophes. *Proc Natl Acad Sci U S A*. 118.

Zanic, M. 2016. Measuring the Effects of Microtubule-Associated Proteins on Microtubule Dynamics in vitro. *Methods Mol Biol*. 1413:47-61.